# Enabling transparent toxicokinetic modeling for public health risk assessment

Sarah E. Davidson-Fritz[1], Caroline L. Ring[2], Marina V. Evans[2], Celia M. Schacht[2,3], Xiaoqing Chang[4], Miyuki Breen[2], Gregory S. Honda[2,3], Elaina Kenyon[2], Matthew W. Linakis[5], Annabel Meade[2,3], Robert G. Pearce[2,3], Mark A. Sfeir[2,3], James P. Sluka[6], Michael J. Devito[2], John F. Wambaugh[2]*

1 Center for Computational Toxicology and Exposure, Office of Research and Development, United States Environmental Protection Agency, Cincinnati, Ohio, United States of America, 2 Center for Computational Toxicology and Exposure, Office of Research and Development, United States Environmental Protection Agency, Research Triangle Park, North Carolina, United States of America, 3 Oak Ridge Institute for Science and Education, Oak Ridge, Tennessee, United States of America, 4 Inotiv, Research Triangle Park, North Carolina, United States of America, 5 Ramboll, Raleigh, North Carolina, United States of America, 6 Department of Intelligent Systems Engineering and Biocomplexity Institute, Indiana University, Bloomington, Indiana, United States of America

* wambaugh.john@epa.gov

## Abstract

Toxicokinetic modeling describes the absorption, distribution, metabolism, and elimination of chemicals by the body. Chemical-specific *in vivo* toxicokinetic data is often unavailable for the thousands of chemicals in commerce. However, predictions from generalized toxicokinetic models allow for extrapolation from *in vitro* toxicological data, obtained via new approach methods (NAMs), to predict *in vivo* human health outcomes and provide key information on chemicals for public health risk assessment. The *httk* R package provides an open-source software tool containing a suite of generalized toxicokinetic models covering various exposure scenarios, a library of chemical-specific data from peer-reviewed high-throughput toxicokinetic (HTTK) studies, and other utility functions to parameterize and evaluate toxicokinetic models. Generalized HTTK models in *httk* use the open-source language MCSim to describe the compartmental and physiologically based toxicokinetics (PBTK). New HTTK models may be integrated into *httk* with a model description code file (C script generated via MCSim) and a model documentation file (R script). *httk* provides a series of functionalities such as model parameterization, *in vivo*-derived data for evaluating model predictions, unit conversion, Monte Carlo simulations for uncertainty propagation and biological variability, and other model utilities. Here, we describe in detail how to add new HTTK models into the *httk* package to leverage its pre-existing data and functionality. As a demonstration, we describe the integration of a gas inhalation PBTK model. The intention of *httk* is to provide a transparent, open-source tool for toxicokinetics, bioinformatics, and public health risk assessment that makes use of publicly available data on more than one thousand chemicals.

**Data availability statement:** The R package httk is available from Figshare at: https://doi.org/10.23645/epacomptox.6062791. v1 R package "httk", including R scripts, data files, and C code, is freely distributed via the Comprehensive R Archive Network (CRAN – https://cran.r-project.org/). The "httk" codebase is provided in a public GitHub repository (https://github.com/USEPA/CompTox-ExpoCast-httk). A change log describing revisions is available at: https://cran.r-project.org/web/packages/httk/news/news.html. HTTK models can be found at: https://github.com/USEPA/CompTox-ExpoCast-httk/tree/main/models.

**Funding:** The United States Environmental Protection Agency (U.S. EPA) through its Office of Research and Development (ORD) funded the research described here. ORD scientists contributed to the research described here. This project was supported by appointments to the Internship/Research Participation Program at ORD and administered by the Oak Ridge Institute for Science and Education through an interagency agreement between the United States Department of Energy and U.S. EPA. JPS acknowledges funding support from the U.S. EPA in grant USEPA RD840027. MWL is a consultant at Ramboll and declares no conflict of interest. XC is an employee of Inotiv, Inc. Her work on this project was funded with federal funds from the National Institute of Environmental Health Sciences, National Institutes of Health under Contract No. HHSN273201500010C.

**Competing interests:** The authors have declared that no competing interests exist.

## Introduction

Decisions on the risk that chemicals may pose to public health involve evaluating its toxicity, exposure, and the dose-response relationship [1]. Next generation risk assessment relies in part on *in vitro-to-in vivo* extrapolation (IVIVE) [2]. Mathematical models describing toxicokinetic (TK) processes predict the relationship between external doses and internal tissue concentrations [3]. Thus, TK models are needed for IVIVE relating bioactive concentrations identified by *in vitro* toxicology experiments to real world doses to which people may be exposed [2,4–6]. While, TK data are needed for IVIVE, human *in vivo* TK data are scarce for many chemicals in commerce and the environment [7].

In high throughput toxicokinetics (HTTK), chemicals are characterized by standardized parameters that can be measured *in vitro* or predicted *in silico* [8]. A series of chemical-independent (that is, generic) toxicokinetic models have been created to use these chemical-specific HTTK parameters [9–11] and enable IVIVE for risk assessment [2,4,12–15]. The generic HTTK models predict the relationship between external chemical exposures and internal tissue concentrations equivalent to bioactive concentrations identified *in vitro*. HTTK enables incorporation of chemical-specific IVIVE into bioinformatic workflows which analyze the toxicological effects across a large number of chemicals, which in turn allows for more rapid chemical risk assessment [15].

The United States Environmental Protection Agency's (U.S. EPA) R package *httk* [9] is one example of an HTTK software tool intended to facilitate the incorporation of toxicokinetic modeling seamlessly into chemical risk analysis workflows. Throughout this document we distinguish the general science of HTTK from the specific implementation of the R package *httk* using capitalization and italics, respectively. The *httk* R package provides a suite of empirical and physiologically based toxicokinetic (PBTK) models [9–11] designed for bioinformatics [15–17]. *httk* is freely distributed via the Comprehensive R Archive Network (CRAN – https://cran.r-project.org/) as a bundled "package" of R scripts, data files, and C code.

Generic PBTK models describe standard physiological processes and a set of standardized parameters that vary between chemicals [8,18]. The chemical specific parameters can often be obtained by *in vitro* or *in silico* methods [4,19]. Since there are many TK processes that are important for only a subset of chemical classes, we expect a generic model to have larger uncertainty than a bespoke chemical-specific model [20]. However, the same generic model may be used across many chemicals, which offers greater confidence in the model implementation and reproducibility, and may also provide a consistent set of predictions across a wide range of chemicals. The key advantage of using generic models is the ability to evaluate the model in the absence of chemical-specific *in vivo* data [21].

Over the past decade, HTTK data has become publicly available for more than a thousand chemicals [8]. Concurrently, public repositories of *in vivo* toxicokinetic observations (that is, chemical concentration in various tissues as a function of dose and time) have become available to allow statistical analysis of HTTK model performance [22]. Typically, we do not expect *in vivo* data to be available for a particular chemical of interest. However, a relevant generic model might be available. Confidence in a generic modeling approach can be estimated by evaluating HTTK predictions across a set of chemicals with *in vivo* data. That is, evaluating a generic HTTK model is not dependent upon the presence of data for a specific chemical of interest, rather its overall predictive performance for chemicals without available data. Any inaccurate assumptions, approximations, omissions, or mistakes in the HTTK model should increase the estimated uncertainty when systematically evaluated across chemicals [21]. The generic nature of HTTK models allow for the estimation of bias, uncertainty, and subsequent correlation of residuals with chemical-specific properties [21,23]. Decision makers can review

the overall predictivity of a generic model of interest and judge whether it can be used for a chemical lacking *in vivo* data but has *in vitro* or *in silico* HTTK values available.

The data and software infrastructure of *httk* provides a platform for potential expansion with new models. Here, we discuss how a new HTTK model may be created and integrated into the *httk* package to use previously evaluated tools and data with the new approach. New models typically simulate important aspects of toxicokinetics currently unavailable in the suite of existing HTTK models in *httk*. Once an HTTK model is developed and integrated with the existing library of chemical-specific HTTK data, it can inform TK applications ranging from regulatory decision making [15,24,25] to systems biology [17,26,27]. For example, inclusion of the inhalation model from Linakis [10] provided a new HTTK model addressing an exposure route previously not implemented.

This manuscript is an appeal to the biological modeling community to help expand the physiological and chemical domains addressed by *httk*. The model development philosophy for *httk* is to create a suite of models, each covering specific exposure scenarios, rather than a single all-encompassing model. Details provided here are intended to aid developers of new models so they may integrate it with *httk* and use its existing data and functionality. Thus, whenever a new model is developed, evaluated, and independently peer-reviewed, the integration process for adding it into EPA's suite of HTTK models is streamlined. While public release is not required, if a new model is integrated into the publicly released version of the *httk* R package that model will be accessible to modelers, bioinformaticians, and risk assessors utilizing the package. Ultimately, the most challenging task is not building a new model, but evaluating its performance [28, 29]. New generic TK models should be statistically evaluated with data in *httk,* which includes a growing library of chemicals. When *in vivo* data are available, each new model incorporated into *httk* should include a comparison of the model predictions relative to the applicable *in vivo* data. An evaluation across the library of HTTK chemical-specific data in *httk* provides an estimate of how generalizable the new model is expected to be. That is, we can establish a chemical domain of applicability for the new model based on correlating chemical features with better or worse predictive ability.

## Results

The Linakis [10] generic gas inhalation PBTK model is used as a case study example to first demonstrate incorporating a new model into *httk*, and then demonstrate evaluating that model for its predictivity. Here, we provide a high-level description of the model incorporation steps and results from the model evaluation with HTTK data for the generic gas inhalation PBTK model. Further, we provide the results of the overall performance of the *httk* R package as a function of time with a series of benchmark statistics, using the new benchmark_httk() function in *httk*. For further details on model incorporation and benchmark performance statistics we refer the reader to "Methods" section.

### Adding a model to *httk*

Table 1 provides step-by-step instructions for adding a new model into the *httk* package environment, allowing it access to all data distributed with *httk*, and potentially enable the model to be distributed to the broader scientific community. The computation time for solving generic HTTK models, which are described by a system of ordinary differential equations (ODEs), in R is substantially slower compared to a compiled language like C. Thus, we opt not to describe generic HTTK models with R code, but instead describe the system of ODEs underpinning a new model with a programming language called MCSim, which can easily convert the model code to C. Once the C file code is generated and added to the R package,

**Table 1. Instructions for adding a model to the httk modeling suite. See Methods for further details, and the supporting information along with Table 2 for examples. Find a user guide on-line at: https://github.com/USEPA/CompTox-ExpoCast-httk/blob/main/models/HTTK-Models-Guide.pdf. Note: [MODEL] is a generic placeholder for the new HTTK model name.**

| | Step Instructions | Comments |
|---|---|---|
| 1 | Describe an HTTK model (parameters, equations, unit conversions) using the MCSim model format | Users can also start by writing their own C script (.c file) to describe the model if they prefer (skipping to Step 3 if doing so), but the MCSim allows declarative rather than procedural modeling and may be more familiar to the PBTK modeling community. |
| 2 | Convert HTTK model from MCSim (.model) file to C (.c file) | Requires download and installation of MCSim https://www.gnu.org/software/mcsim/ or MCSim_under_R (https://github.com/nanhung/MCSim_under_R). |
| 2a | • Run building_mod.R file once (in MCSim_under_R> my_R_project folder) | • This step generates the mod.exe file required to convert the.model file to the.c file |
| 2b | • Run one of the script_compile_run.R files (in MCSim_under_R> my_R_project folder) | • This step will generate multiple files", but the only important one for *httk* is the.c file. For example, "model_gas_pbtk.c". We suggest copying the output file and adding "-raw" to the end of the unedited file name, before the.c extension, to be consistent with naming conventions set in this paper (for example, "model_gas_pbtk-raw.c"). |
| 3 | Add C Version of Model to R | |
| 3a | • Change the function names within the.c file | • To be consistent with the naming convention, try to name the function, "[Function]_[MODEL]". |
| 3b | • Copy the formatted.c file into the "httk/src" folder | • This is the model file that will be compiled into the *httk* package when it is built. Remove "-raw" from the file name if you've already added it |
| 3c | • Update the init.c file | • The init.c file lists all the functions *deSolve* needs for all the models in one place. |
| 4 | Describe the Model within R | • Generate/Modify the modelinfo_[MODEL].R file that adds information to the global model.list describing how other core *httk* functions should interact with each model. |
| 5 | Using Core Functionality | |
| 5a | • Create parameterize_[MODEL].R file | • The function generating all necessary model parameters, which are specified within the modelinfo_[MODEL].R file. This can be reused from other models but if new parameters are needed a new function should be created. |
| 5b | • Create solve_[MODEL].R file | • We create a model specific wrapper function that internally calls the solve_model() function with the arguments that would be typical for the model (such as specific dosing routes). |
| 6 | Rebuilding and installing a new version of *httk* | To use the new model, the revised *httk* package must be rebuilt and installed in R. We recommend changing the version number in the DESCRIPTION file to reduce confusion. |

we create a modelinfo_[MODEL].R file describing how the core *httk* functions should interact with the new model. Additional functions to use various *httk* extensions such as Monte Carlo uncertainty and variability analysis may then be created. Finally, once a model is added, we must install the updated version of *httk* to utilize the new model. The Methods section "Creating and integrating a new HTTK model" walks through all these steps in detail. In this section, we will focus on the case study example and providing details about how we integrated the inhalation gas PBTK model from Linakis [10] to *httk*.

In Linakis [10] a generic gas inhalation PBTK model (gas_pbtk) was added to *httk*. The files required for adding this model are included as supporting information (see S3–S8 Files). The model was developed through combining a pre-existing generic inhalation model of Jongeneelen and Berge [30] with the inhalation model of Clewell III [31], which noted that some chemicals may be absorbed into the mucus or otherwise trapped in the upper respiratory tract. Then the steps of Table 1 were followed: First, a model description was written in MCSim (S3 File; "model_gas_pbtk.model") including state variables, parameters, allometric scaling and differential equations. The model file was translated to a compilable C script with the "mod" function in MCSim (S4 File; "model_gas_pbtk-raw.c"). The "raw" C file was then edited to include the necessary updates for integration with other models, data, and tools already present in *httk*. For example, MCSim gives default names to the functions in the auto-generated C script which must be renamed to avoid duplicative function names potentially overlapping with other models in the *httk* model suite. Once the C script was formatted to have unique function names

it was saved without the "raw" suffix (S5 File; "model_gas_pbtk.c"). The final step for the model code was adding the formatted C script into the "httk/src" sub-directory and modifying the "httk/src/init.c" file to list the newly created functions in the new HTTK model C script.

To allow interoperability between the gas_pbtk model and the core HTTK functions in *httk,* a model information (or "modelinfo") file was created (S6 File; "modelinfo_gas_pbtk.R"). Beyond basic definitions for the relevant functions, the modelinfo file describes the default units and state of matter ("gas" or "liquid") for each compartment in the model. For the gas inhalation model these differed (for example, "exhaled breath" vs. "venous plasma"). Additionally, two new R functions were created for parameterizing the model and interfacing with the model solver (S7 and S8 Files; "parameterize_gas_pbtk.R" and "solve_gas_pbtk.R", respectively). For the gas_pbtk model, the parameterization function was adapted from the parameterize_pbtk() function code. In this case, the code for the generic pbtk model was copied and the equations for the lung were modified to use forms similar to Jongeneelen and Berge [30], with the addition of a mucus/upper respiratory compartment from Clewell III [31]. However, one may also use the parameterize_pbtk() function itself to generate parameters for a variety of tissue lumping schemes. For example, Kapraun [11] called parameterize_pbtk() twice within the parameterize_fetal_pbtk() function – once for the mother and once for the fetus. The three R scripts defining gas_pbtk specific functions (that is, those files with the ".R" extension mentioned earlier in this section and step 4 of Table 2) were added to the "httk/R" sub-directory. Finally, the *httk* package was recompiled – as described in the "Methods" section. This new PBTK model was included with the public release of *httk* (v2.0.0) [10].

Of the 1046 chemicals with *in vitro* measured data, only 962 are considered non-volatile or semi-volatile, and therefore within the applicability domain of the default "pbtk" model. The "gas_pbtk" model is applicable to all 1046. Linakis [10] added data for several dozen volatile chemicals to the internal *httk* data tables, and these data were also distributed with *httk* v2.0.0 and subsequent versions.

**Table 2. Steps taken to add the gas inhalation model to httk following the step-by-step instructions outlined in Table 1.**

| | Model Adding Step | Supporting Information (Filename Expected by *httk)* | Specifics |
|---|---|---|---|
| 1 | Use a declarative language (MCSim) to describe the ODEs for a PBTK model with inhalation. | S3_MCSim_example.model (model_gas_pbtk.model) | Started with the basic *httk* PBTK model (model_pbtk.model) and added equations from Jongeneelen and Berge [30] and Clewell III [31]. |
| 2 | Use the MCSim function mod.exe to create a compilable C script (.c file) from the.model description, and format the C script to be compatible within the *httk* model suite. | S4_raw_C_example.c (model_gas_pbtk-raw.c) | MCSim autogenerates the same function names, so we rename derivs, initmod, getParms (among others) to allow *httk* to load multiple models at the same time. For example: derivs_gas_pbtk, initmod_gas_pbtk, getParms_gas_pbtk |
| 3 | Add C code for new model to the *httk* R package. | S5_C_for_httk_example.c (model_gas_pbtk.c) | The formatted C script (.c file) is placed in the "httk/src" directory where it can be compiled by R when the package is built. |
| 4 | Connecting the new model with core *httk* functionality. | S6_modelinfo_example.R (modelinfo_gas_pbtk.R) S7_param_func_example.R (parameterize_gas_pbtk.R) S8_solve_model_wrapper_example.R (solve_gas_pbtk.R) | We needed to calculate a new parameter, the blood to water partition coefficient, as part of our parameterization procedure. This meant that we needed the Henry's law constant for each chemical. |
| | Files not added | (calc_analytic_css_gas_pbtk.R) | Because a function was not available to calculate steady-state concentration, we did not integrate this model with the HTTK Monte Carlo functionality including HTTK-Pop. |
| 5 | Install the new version of the *httk* R package. | | The inhalation model was first included with v2.0.0. |

In Linakis [10] a steady state solution was not developed. Without a steady state solution Monte Carlo simulation are computationally expensive and precludes many of the toxicological IVIVE tools built into *httk* (such as calc_mc_oral_equivalent()). In the absence of a steady state solution, the function calc_mc_tk() will run a series of full TK models and report the mean and standard deviation of the concentration at each time point requested.

We have evaluated the generic model on two levels. In the next section, we re-evaluate the predictive accuracy for the Linakis [10] inhalation model. This demonstrates the model performance evaluation process for new HTTK scenarios. After that, we will then evaluate any unintended impact of adding the new model against the historical performance with a set of key HTTK performance statistics.

## Evaluating a new model

Formal inclusion of a new model into the CRAN distributable version of *httk* requires that it undergoes model evaluation and an independent peer-reviewed prior to inclusion. Model evaluation is often the most difficult part, as it requires compilation of data amenable to perform the evaluation – for example, *in vivo* observations for chemicals relevant to the scenario explored by the model. Sometimes these data do not exist but, where they do exist, statistics indicative of a model's predictive ability, such as the root mean squared error on the $\log_{10}$ scale (RMSLE), absolute fold error, and the coefficient of determination (that is, $R^2$), should be estimated. For $R^2$ we suggest using the total explained variance (predicted vs. observed) rather than the $R^2$ for a regression of the observations on predictions. There is no prescribed threshold of accuracy for inclusion to *httk*, though a case must be made that resulting model predictions are fit for purpose. Performance metrics for model predictability also serve as benchmarks whenever functions within the R package are updated and/or new data are added. We anticipate a general trend toward more accurate model predictions, though this may not always occur.

Linakis [10] demonstrated an *httk*-based approach for evaluating generic TK models across multiple chemicals using *in vivo* measured data. The RMSLE was estimated for model predictions on 41 volatile organic chemicals across 118 experimental inhalation exposure conditions in two species drawn from the Chemical concentration vs. time database (CvTdb) [22]. Fig 1 was obtained by running the RMarkdown vignette "Linakis et al. (2020): High Throughput Inhalation Model" to re-generate Fig 2 from Linakis [10] with the new features of *httk* (S9 File). It should be noted, the original figure in that work compares observed chemical concentrations with predictions on a logarithmic scale. Three regressions were performed: separately for human and rat data, and then combined for an "overall" trend. In that study, prediction errors were analyzed across all chemicals jointly, so that chemicals with more data had greater weight. Since the available data can greatly vary between chemicals, we generally recommend first averaging per chemical, then averaging across chemicals, but given the aim here was to reproduce the previous work we do not do this here.

Changes in results between our Fig 1 and the original [10] figure demonstrates the impact of incorporating new features within *httk* on the inhalation PBTK model. Here we observe that the RMSLE has increased from 1.11 to 1.19. This discrepancy may be the result of several updates including: 1) restricting the model chemical domain to non per- and polyfluorinated alkyl substances (non-PFAS), 2) refinement of physico-chemical properties, and 3) on-going curation and refinement of the *in vivo* data within CvTdb. The R package *deSolve* [32] takes advantage of the computational efficiency of compiled C code ("httk/src/model_gas_pbtk.c") to calculate the derivative of the model ODEs and obtain rapid analytic solutions. The indices for the equations (indicating which variable is which) must match within both the C code and the R code. Within the modelinfo_gas_pbtk.R file the variable derivative.output.names lists

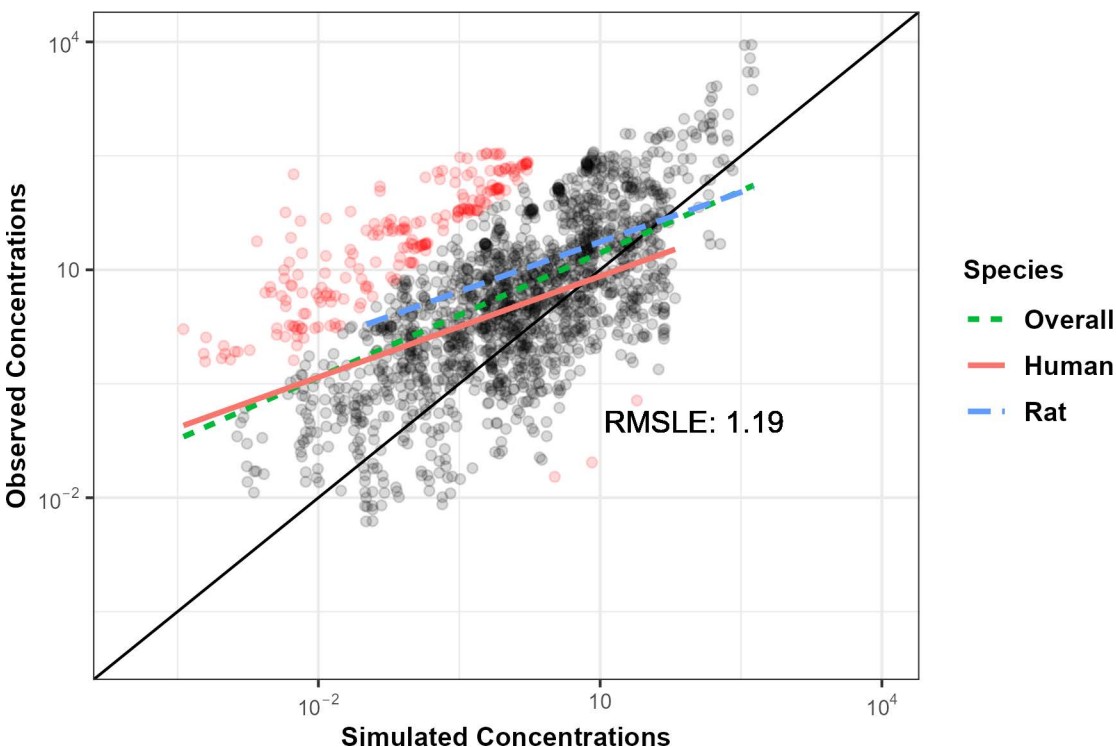

**Fig 1. Updated evaluation of Linakis [ 10] inhalation model predictivity.** Updated evaluation of [10] inhalation model using data from 118 *in vivo* experimental conditions in 2 species on 41 volatile organic chemicals obtained from CvTdb [22]. Black points indicate predictions within ten-fold of the observations, red points indicate more than hundred-fold difference. Line dashing indicates regressions between observations and predictions for humans (solid, red), rats (long dash, blue), and overall (short dash, green).

the expected outputs and must do so in the same order as the C code. When developing the model initially, we mislabeled the variables (that is, the wrong order) which provided different units for the respective variable (for example, exhaled breath concentration is provided in both ppmv and µ M). This mistake caused the RMSLE to increase to nearly 1.8.

## Assessing any impact on the rest of the package

The new function benchmark_httk() performs a series of checks and estimates performance statistics for assessing the state of the package. For example, benchmark_httk() reports the number of chemicals with available data (which should not change unexpectedly). An additional sanity check is that outputs are generated for units of both mg/L and µ M, and then the ratio $C_{mg/L} \Big/ C_{\mu M} \times \dfrac{1}{1000 \times Molecular\,Weight}$ is calculated (this should be 1). The benchmark_httk() function also provides predictive performance benchmarks to assess the impact of changes in data, models, and feature implementations within *httk*. In the past, some code changes made for one feature of *httk* have unintentionally impacted others. Most notably were unit conversion errors introduced with hard-coded calculations (v1.9, v2.1.0). Package evaluation with this new benchmarking tool is meant to identify errors prior to version release, thus reducing the impact on future users. For further details on the checks and performance statistics conducted with benchmark_httk() we refer the reader to the Methods section "Benchmarking *httk*".

Performance of older *httk* versions were retroactively evaluated by manually installing previous versions of the package from https://cran.r-project.org/src/contrib/Archive/httk/, then adding the code for benchmark_httk() function at the command line interface and calculating the various benchmark statistics. Fig 2 shows the RMSLE to evaluate the concordance of several common HTTK statistics – including three IVIVE statistics, two Monte Carlo tests, and a partition coefficient test – which serve as benchmarking metrics for the package performance overall.

The *in vivo* statistics in benchmark_httk() establish the overall predictivity for area under the concentration vs time curve (AUC), peak plasma concentration ($C_{max}$), and concentration at steady state ($C_{ss}$) in *httk*. The *in vivo* statistics are currently based on comparisons to the *in vivo* data compiled by Wambaugh [33]. A summary of changes to *httk* by version are available at: https://cran.r-project.org/web/packages/httk/news/news.html.

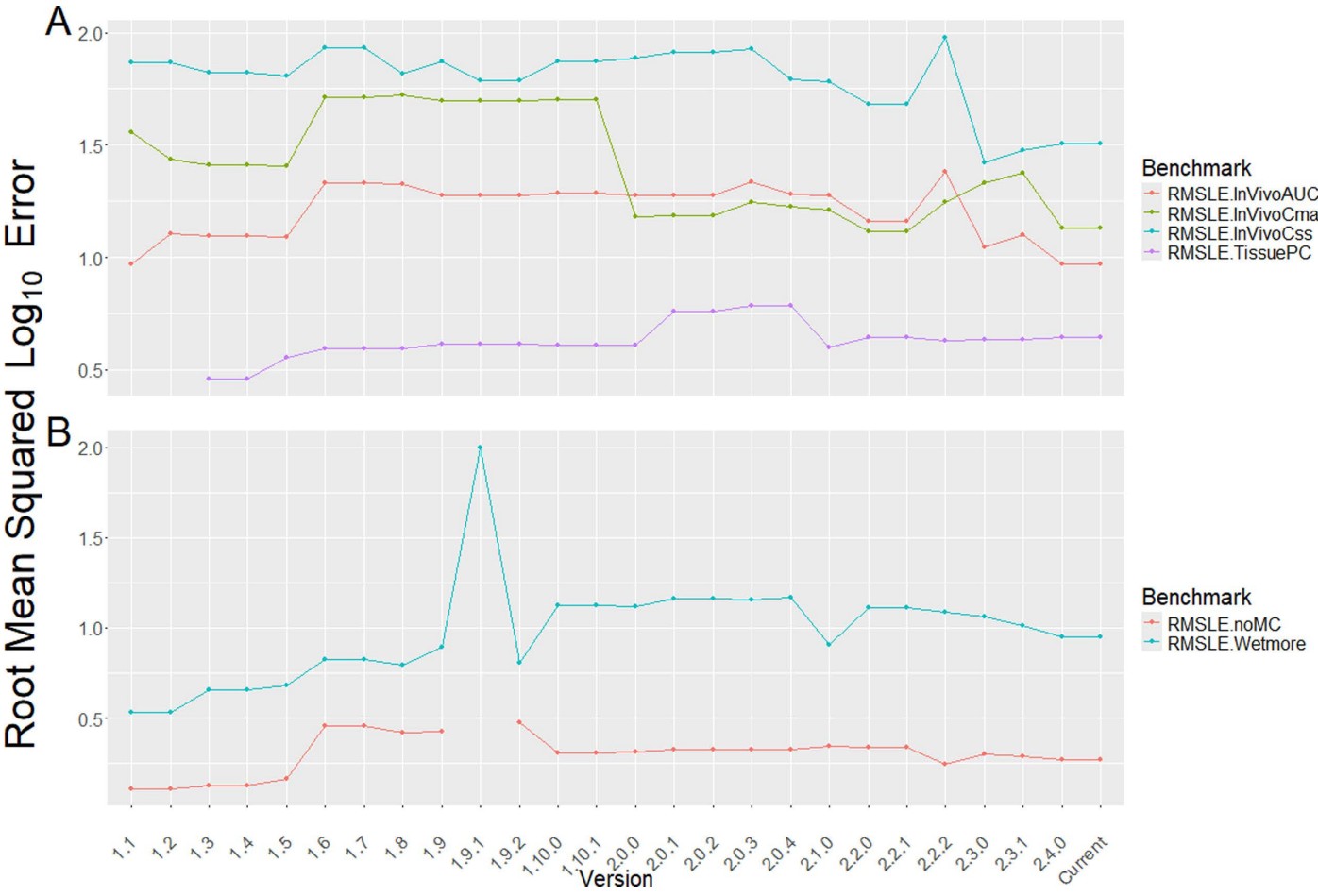

**Fig 2. Benchmarking statistics plot evaluating *httk* versions over time.** Panel A: Output generated by benchmark_httk() for both past and current performance of *httk* versions for a series of tests. Here performance is characterized by the root mean squared error on the $\log_{10}$ scale (RMSLE). IVIVE Statistics: *in vivo* area under the plasma concentration vs time curve (AUC), maximum observed plasma concentration (Cmax), and the plasma concentration at steady state (Css) are all based on data collected by Wambaugh [33]. "Tissue PC" compares predicted tissue partition coefficients with literature *in vivo* estimates collected by Pearce [34]. Panel B: Wambaugh [33] Monte Carlo Tests: the "noMC" RMSLE benchmark compares the median of the Monte Carlo $C_{ss}$ distribution, obtained with calc_mc_css(), with the $C_{ss}$ predictions of calc_analytic_css(). The "Wetmore" RMSLE compares the 95th percentile predictions from calc_mc_css() with the estimates from SimCyp [18] as reported by Wetmore [12], Wetmore [13].

The partition coefficient test ("TissuePC") provides an important check on the *httk* implementation of the Schmitt [35] model tissue to plasma equilibrium distribution. Pearce [34] collected equilibrium tissue:plasma concentration ratios across a range of diverse chemicals and tissues. These predictions heavily rely on accurate descriptions of tissue composition and the ability to predict the ionization state of compounds being modeled. Therefore, the ability to predict tissue partitioning is particularly sensitive to changes in how physico-chemical properties are modeled by *httk*.

Monte Carlo benchmarks include the Wetmore and noMC tests. The "Wetmore" benchmark compares the calc_mc_css() Monte Carlo 95th percentile steady state plasma concentrations for a 1 mg/kg/day exposure against the steady state values calculated by SimCyp [18] and reported in Wetmore [12], Wetmore [13]. These have gradually diverged as the assumptions for *httk* have gradually shifted to better describe non-pharmaceutical commercial chemicals [23]. The "noMC" benchmark compares the median of the distribution from calc_mc_css() against the non-varied steady state result from calc_analytic_css(). In both cases, deviation between the two values is expected but should not deviate wildly unless major changes have been made to *httk*'s Monte Carlo sampler or the underlying data. Notably in Fig 2, panel B, a unit conversion bug was introduced in the code for *httk* v1.9.1, altering several key results. The benchmark_httk() function was not available at the time, but with its implementation errors like this one can be identified and corrected prior to new version releases.

Both new models as well as other changes to the underlying *httk* code can generally be benchmarked against existing *in vivo* data using benchmark_httk(). Developers of new models may want to create and use *in vivo* data sets to benchmark new models against more relevant data (for example, characterizing new routes of exposure).

## Discussion

Assessing the risk posed by tens of thousands of non-pharmaceutical chemicals that may occur in the environment and commerce to public health is necessary [36]. For ethical reasons it is typically infeasible to obtain human toxicokinetic data from controlled exposure(s) to non-therapeutic chemicals. Controlled exposures are where the magnitude, route, and timing of a chemical exposure are sufficiently known to allow mathematical modeling. Furthermore, there are limited resources for obtaining similar data in animals, especially considering the number of compounds for which a data is needed is vast. Thus, there is a general lack of *in vivo* data for building and evaluating toxicokinetic models for non-pharmaceuticals [7,12]. In these cases, generic PBTK models offer an alternative path [18,30,37–40].

PBTK models are widely used within the pharmaceutical industry to better understand drug absorption and disposition [41, 42]. Evaluating the correspondence between predictions and observations allows for the estimation of bias and uncertainty. Decision makers may then consider using models to extrapolate to other situations (for example, dose, route, and physiology) where data may be unavailable [43]. In non-pharmaceutical chemical risk assessments, model reproducibility, statistical evaluation [44], and data availability are all barriers to the incorporation of PBTK models into public health risk assessment [45]. Despite the deterministic nature of scientific computing (even including reproducible pseudo-random number generation [46]) it is often difficult to reproduce computational results [47]. Problems with model reproducibility are due, in part, to the lack of standardized means to distribute, document, and evaluate models [48]. Furthermore, discrepancies emerge between initially conducting research and the final presentation of analyses in a research manuscript [49]. These limitations are due, in part, to the mechanisms available for describing models.

HTTK methodology predicts human doses with the potential to cause bioactive concentrations estimated from *in vitro* high throughput screening assays (that is, quantitative IVIVE [50]). One goal of the *httk* R package project is to add new HTTK models exploring various factors that may impact exposure. For example, new HTTK models may enable evaluation of a chemical exposure for various exposure scenarios, physiological processes, or non-terrestrial species (such as fish) to better perform *in vitro* to *in vivo* extrapolation. The *httk* package is intended to maximize the reuse of existing *in vitro* measurements of TK determinants, physico-chemical property predictions, and functions to perform routine calculations with those values. By harmonizing functions and conserving data across HTTK models we aim to increase the verification of data and models. We also aim to improve package functionality, while improving transparency, reproducibility, and the validity of new models.

The *httk* R package collects open-source data and models to make HTTK methods both available for chemical risk evaluation and accessible for review and refinement by the scientific community. The U.S. EPA's CompTox Chemicals Dashboard [51], the United States National Toxicology Program's Integrated Chemical Environment tool [52], the European Food Safety Authority's TK-Plate [53,54], Health Canada's science approach document "Bioactivity Exposure Ratio: Application in Priority Setting and Risk Assessment" science approach document [24], and Certara's SimRFlow for SimCyp [55,56] all make use of the existing data and models from *httk*. Making the data and simulation functionality from *httk* more accessible to new models will hopefully encourage the modeling community to develop new HTTK models and expand IVIVE.

The primary intention of HTTK modeling is for high-throughput human health risk prioritization. These models must be fully evaluated to consider their use in public health decision making [44,57–60]. Clark [45] identified six areas to assess any PBTK model intended for public health risk assessment: 1) purpose, 2) structure and biology, 3) mathematical descriptions, 4) computer implementation, 5) parameter values and model fitness, and 6) any specialized applications. Ideally, when new tissues or physiologies are added to the *httk* modeling suite data characterizing these aspects will be used and made available to the modeling community. In the absence of such data, models for new tissues should at a minimum examine whether the predictive performance has degraded for other tissues (such as blood) where data are available.

The *httk* R package facilitates systematic statistical evaluation using *in vivo* data on a range of chemicals. This evaluation allows the quantification of uncertainty and characterization of confidence for decision making. Using generic models in a standardized environment, such as *httk*, provides a variety of well-defined physiologies with specific routes of exposure that may be evaluated for many chemicals. Thus, *httk* addresses the first four points of outlined by Clark [45]. Matters of parameter appropriateness and model fitness require further evaluation using tools such as statistical regression and machine learning [29]. The R statistical programming environment provides many methods for these more in-depth model evaluations. Databases, like CvTdb [22], provide *in vivo* data to compare with model predictions for model evaluation. Finally, the chemical domain of applicability may often be estimated empirically by correlating the model performance with chemical properties [61].

Generic models typically describe a single physiology and characterize chemicals with a set of standard descriptors. Thus, we only expect a generic model to perform well for questions relevant to the physiology included and chemicals well-characterized by the descriptors that are measured. If a key process or property to a chemical class is omitted, we expect the model to perform poorly for that particular class [21,23]. While larger uncertainty is expected for any one chemical, a generic model allows greater confidence in model implementation [21,62] and

a better understanding of the overall trends across compounds. Using a model based on *in vitro* data for chemicals without *in vivo* data may be considered appropriate if the magnitude of uncertainty and chemical properties are found to be statistically correlated [21].

Sophisticated computer programmers are not necessarily expert biological modelers, and vice versa. There are significant computational advantages to using lower-level model implementations. For example, languages like C or Fortran can be compiled into assembly machine instructions that are more computationally efficient than interpreted languages like R. However, implementing biological models in compilable languages can be tricky. Biological modelers are often more comfortable with declarative modeling languages, which are more human interpretable (such as Advanced Continuous Simulation Language (ACSL) [63] or Berkeley Madonna [64]) [65]. Declarative languages for models define the variables and describe what must be true about the relationships between them as well as how they change over time. A declarative model description does not have to be written in a specific order. Automated tools can then convert the declarative model description into more efficient code by organizing the instructions into the stricter order required by a lower-level (but faster) computer programming language. Therefore, it is of great use that open source tools like MCSim [66], which provide the ability to translate a descriptive language for models (that is, MCSim) into a rapid, compilable language like C.

The "software ecosystem" [67] design of *httk* intended is to allow reuse [68] of data and functions (that is, recurrent calculations) from peer-reviewed scientific literature. As with any R package [69], reusable functions make *httk* more modular [70] and better enable model developers to integrate new models and conduct consistent analyses. Developers may also extend existing functions to meet their needs. Function modularity provides "scoping", which helps to limit the unintended impacts of new features and updates from developers on existing functionality [71]. Additionally, modularity centralizes code updates to the fewest places possible when errors, that is computational/code "bugs", occur in recurring calculations. Though these practices allow flexibility while attempting to prevent new contributions from breaking existing functionalities, bugs in the code will inevitably occur. Therefore, it is critical that developers are able to conduct method and model evaluations to provide indications of potential bugs and help identify and diagnose these issues which may be unidentified by other means [59].

While there is no reason HTTK data and models must be implemented in R, we are unaware of comparable tools in other, similar languages such as Python. It is certainly possible to use MCSim with Python [72] and to use the R packages, like *httk*, through Python using rpy2 [73] for Python [74]. However, as far as we know the only open-source database of chemical-specific HTTK parameters and models is the R package *httk*. Furthermore, the Python package ctx-python (https://github.com/USEPA/ctx-python) and R package ctxR [75] allow for the retrieval of pre-computed predictions made with *httk* from the CompTox Chemicals Dashboard [51].

In Table 1 and the "Methods" section, we describe how new models may be created and integrated with *httk*, and how previously evaluated tools and data may be used with new HTTK modeling approaches. Ultimately, the challenge is not building a new model, but rather evaluating the new model for its ability to provide accurate predictions under new conditions (for example, different exposure routes). New generic HTTK models can and should be statistically evaluated across the growing library of chemicals included in *httk*. Each new HTTK model added to the *httk* package should include an evaluation of model predictions relative to available *in vivo* data. Decision makers can then consider whether the generic model predictions based on *in vitro* data can be used to extrapolate internal doses for chemicals lacking *in vivo* data.

Though we can integrate new models into *httk*, there are still some limitations to the current tools provided in the R package. For example, a feature to check the mass balance of model prediction data, which is output by solve_model(), does not currently exist. Other future development directions include establishing methods for model operations (that is maintaining and evaluating HTTK models), addition of new datasets, streamlining access to data by functions within *httk*, improving the implementation of Monte Carlo simulation for greater flexibility when extending a new model to have uncertainty/variability propagation, and adding functions to continually standardize and streamline the process of adding new generic models to the *httk* R package.

The European Union Registration, Evaluation, Authorisation and Restriction of Chemicals has already begun the transition away from *in vivo* testing [76]. EPA has committed to reducing and replacing animal testing, with near term milestones articulated in the EPA NAMs Work Plan [77,78]. *In vitro* new approach methodologies (NAMs) will be a cornerstone of next generation risk assessment [79]. Placing the results of NAMs in a human health context will require toxicological IVIVE for thousands of chemicals. This is the role for which *httk* is being developed [80].

## Methods

Reynolds and Acock [81] wrote that "modularity and genericness open models to contributions from many authors, facilitate the comparison of alternative hypotheses, and extend the life and utility of simulation models." The same intention motivates the development of the *httk* R package for high throughput toxicokinetics. Pearce [9] described the original release of *httk* which comprised an HTTK model suite including a one- and three-compartment empirical TK model along with a simplified PBTK model. Ring [82] and Breen [83] focused on the Monte Carlo human variability simulator (namely, "HTTK-Pop"). Wambaugh [14] discussed how the Monte Carlo approach also allowed propagation of chemical-specific measurement uncertainty. *httk* uses the built-in R documentation functions, which currently provide substantive detail at the level of individual functions and data sets. However, the current documentation does not provide a clear description of how a new HTTK model may be incorporated into *httk* to utilize the existing tools and data. In the following sections, we describe the basic process for integrating a new HTTK model into the package, as well as provide information on how to leverage some of the other existing tools and available data.

Table 1 provides generic set of step-by-step instructions for adding a new model into the *httk* environment, which then enables the new model to access all data distributed with the package, and hopefully allows the model itself to be distributed to the scientific community. To ensure accurate description and evaluation of the models, *httk* developers use computer programming and modeling best practices such as version control software, unit tests, generalizable functions for creating a "software ecosystem", and model evaluations. U.S. EPA uses the version control software Git [84] and maintains the codebase in a public GitHub repository (https://github.com/USEPA/CompTox-ExpoCast-httk). Git provides transparent code tracking, including a history of changes over time and versions. Unit tests are part of general best practices when developing an R package [69,85] and help determine if changes to one feature of *httk* may unintentionally introduce code bugs or impact others by observing discrepancies between a set of expected and actual results (see the "httk/tests" sub-directory for existing unit tests).

Each of the following sub-sections on methods describes in detail each element necessary for model developers to know when integrating their generic HTTK models to use the modularity of *httk*. Additionally, we outline and describe some of the tools that aid in the

integration process, including core *httk* functions, data, and other features (for example, model extensions like steady-state modeling). Though we mention specific functions, data, and features available in *httk*, this manuscript is not intended to provide an exhaustive list or description of each *httk* functionality. Rather, the specific tools mentioned are meant to demonstrate the utilization of key functionalities in *httk* for model integration and/or extension. For further details on functions, data, features, etc., we refer the reader to the 'help' files – which can be accessed via the help() function – vignettes, and other R resources to navigate the *httk* directory structure [69]. As with any R package, specific components of the package (for example, functions and data) may be used without loading the entire package by adding the prefix "httk::" when developing applications or conducting analyses in R.

### Creating and integrating a new HTTK model

Adding a new model to *httk* relies upon the use of three software programs/coding languages – namely MCSim, C, and R. Fig 3 provides a general schematic of the necessary files and major components connecting the various software used to solve a typical HTTK model.

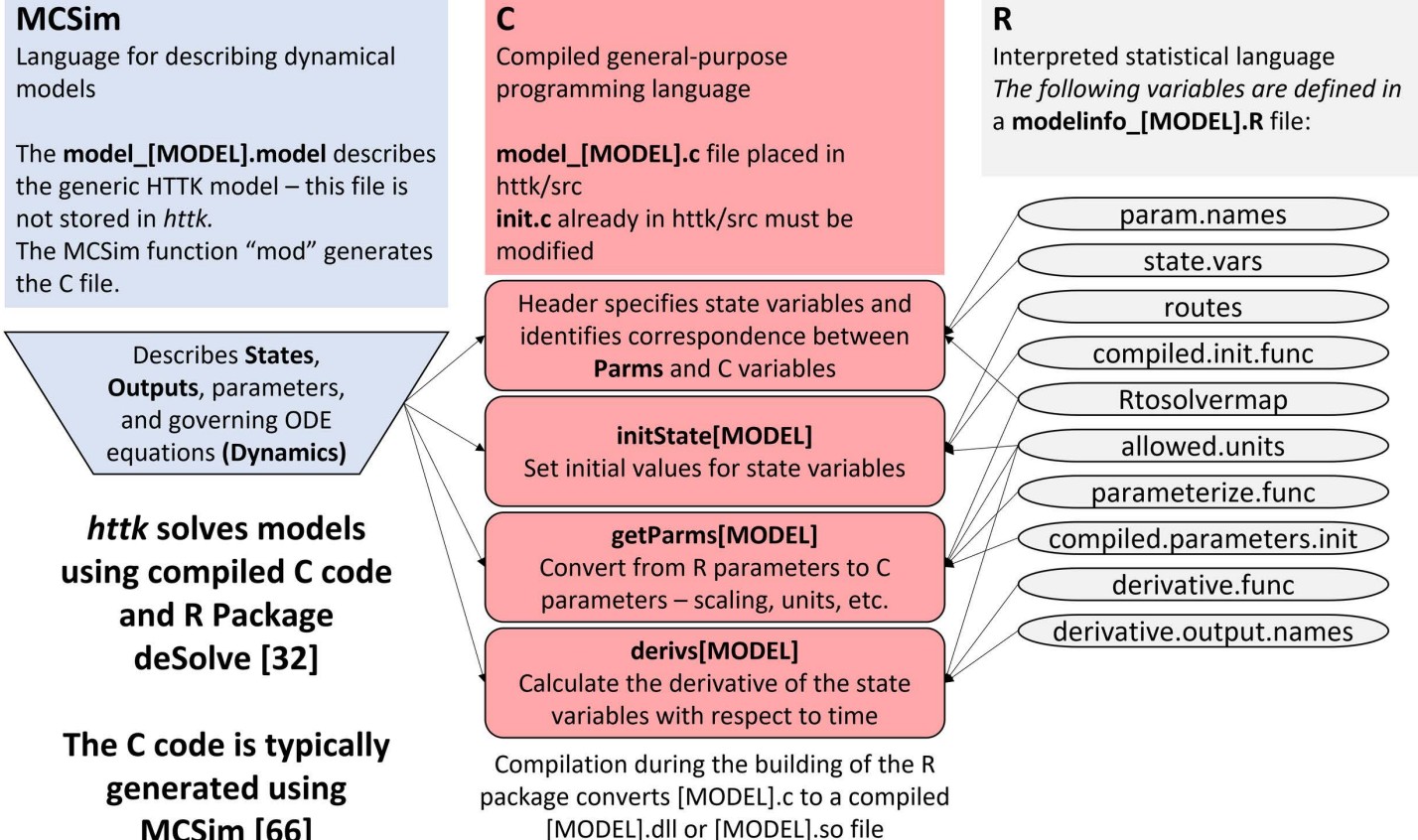

**Fig 3. Schematic of HTTK model files and connections.** General schematic of the three software programs used to generate and implement a new generic HTTK model with the respective files, functions, and objects required. The MCSim file, namely model_[MODEL].model, describes the ODE for the new HTTK model. The C file (updated), namely model_[MODEL].c, can be compiled so that *deSolve* can rapidly calculate the evolution of the state variables with time. The model information R script, namely modelinfo_[MODEL].R, describes how the model should interact with *httk* core functionality.

## Describe the TK model in MCSim

The first step in Table 1 is to create a script that defines the set of equations in the ODE model for the time-evolution of the toxicokinetic model of interest. Here, we assumed the modeler uses MCSim [86], a descriptive dynamical modeling language, to define the HTTK model (model_[MODEL].model).

MCSim was originally designed to implement mathematical models for various purposes including MCMC parameter inference. For *httk* we only take advantage of the model descriptive language because it allows someone trained in biological modeling to explicitly define what must be true about a model without necessitating programming expertise in compliable languages, like C. Alternatively, modelers with experience developing models in the coding language C may write their HTTK model definition file in a C script (model_[MODEL].c) and save it in the "httk/src" subdirectory of the *httk* package. In the latter scenario, this section may include some considerations when writing the model description file in C but can otherwise be skipped.

*httk* uses MCSim to describe some, but not all aspects of the simulation. In particular, dosing (including initial conditions and later events) are handled within R. Table 3 provides the list of key components of an ODE model for HTTK, which are necessary for appropriately defining and incorporating the model, and their mapping between MCSim (.model file), the C code (.c file), the *httk* R package (modelinfo_[MODEL].R file), and the *deSolve* package (that is, the ode() function call within the solve_model() functions) [32,87]. To describe a model so that it can be integrated with *httk* we must consider 1) the System of Differential Equations, 2) States, 3) Times, 4) Events, 5) Inputs/ Forcings, and 6) Outputs. *httk* uses MCSim to describe some, but not all aspects of the simulation – particularly dosing (including initial conditions and later events) which is handled within R.

The ODE system describes the evolution of state variables with time as a function of the current state and model parameters. Specifically, we provide calculating the time derivative of the model ODEs. For *httk*, the deSolve::ode() function argument "func" is set to the name of a compiled function that returns a vector of derivatives for the system of ODEs describing the model. The name of the derivative function is specified by variable "derivative.func" in the modelinfo_[MODEL].R file.

The state variables are those whose time evolution are described by the ODEs, and "Model Parameters" are the non-state variables. Equations, in the ODE system, depend on many parameters describing both physiological and chemical properties. However, there is no explicit "Parameters" block in MCSim, rather they are collected into a list based on wherever they show up. Time points for simulation include timing of events (such as dose administration) and necessary predictions (such as measurements). For *httk*, predictions are only made for those times points which are explicitly specified in the "times" argument, while the solver and the dose regimen may add other times.

Events are instantaneous changes in state, such as an additional exposure to a chemical. Events are handled by breaking the simulation into the time before and after the change. *deSolve* uses numerical integration algorithms to describe continuous changes in between events. Events impose discontinuities in state variables at specific times. On the other hand, Inputs/ Forcings are changes in other external factors that are not state variables (for example, chemical concentration in air or constant infusion dosing). Inputs are independent variables for the model ODE system. An R data.frame, called "eventdata" (which is determined from the dosing.matrix, doses.per.day, and daily.dose if available), is constructed by the httk::solve_model() function and passed to the deSolve::ode() function argument "event".

**Table 3. Major components for describing an HTTK model between MCSim, deSolve, and httk.**

| ODE Components | MCSim | C Code ("httk/src") | R Code ("httk/R") | *deSolve* |
|---|---|---|---|---|
| **System of Differential Equations:** The ODE system describes the evolution with time of the state variables as a function of the current state and model parameters. | MCSim includes a specific "Dynamics" block for specifying model equations. In MCSim, the order does not matter. | MCSim function "mod" translates the ODEs to C code for calculating the derivative. For C, the values evolve as each line of code is executed. | modelinfo_[MODEL].R points to a function "derivative.func" that calculates the derivative of the ODEs. | The function deSolve::ode() uses a compiled function that returns a vector of derivatives that is specified by variable "derivative.func" in the modelinfo_[MODEL].R file. |
| **States:** The state variables are those described by the system of ordinary differential equations (ODEs). | MCSim includes a specific "States" block. | C arrays "y" and "ydot" contain the state variables and the derivative with respect to time. | modelinfo_[MODEL].R describes "state.vars" which should be in the order of derivatives returned from "derivative.func". | The first argument to function deSolve::ode(), both defines initial values for state variables and implicitly the number of state variables. |
| **Model Parameters:** Each equation describing the time-evolution of a state variable may contain multiple parameters describing both physiological and chemical properties. | List of parameters, named and assigned values where necessary in the MCSim code. | C array "parms" contains all N parameters used by the model, the first value is parms[0] and the last is parms[N-1]. | In modelinfo_[MODEL].R, "Rtosolvermap" identifies which R parameters link to C parameters described by "compiled.param.names". These must be defined in the order used by C. | deSolve::ode() takes an argument "parms" containing just the parameters needed by the derivative function "func". |
| **Times:** Time points for simulation, including timing of events (such as dose administration) and necessary prediction (such as measurements). | Simulation times do not need to be explicitly described for our use of MCSim. | The derivative function is written to calculate the instantaneous derivative at time "pdTime". | The "times" vector only requires the times to be observed or predicted – dosing and intermediate times are added automatically. | The deSolve::ode() function argument "times" takes a vector of times where predictions are needed, though intermediated times are calculated. The first value must be the initial time. |
| **Events:** Instantaneous changes in state, such as an additional exposure to a chemical. The simulation is divided into the time before and after each event. | We do not make explicit use of MCSim's Initialize or Inputs block, and instead use deSolve::ode(). | Events are handled outside the C code by *deSolve*. The C code describes the model between events. | An R data.frame, called "eventdata", is constructed by httk::solve_model() and passed to the deSolve::ode() function argument "event". | At each event one or more state variables are changed instantaneously [32]. The R vector "y" defines initial values for state variables. |
| **Inputs/Forcings:** Changes in other external factors (for example, chemical concentration in air or constant infusion dosing). | MCSim includes a specific "Inputs" block. Inputs may change over time due to Events or Forcings [32]. | The C array "forc" contains the values of the forcing variables. | Determined from initforc, forcings, and fcontrol if available. | The forcing function is a time series of constraints – the relevant state variables must match the forcing [32]. |
| **Outputs:** The ODE model components to export from the model evaluation. Changes in state variables must be returned. | MCSim includes a specific "Outputs" block. They can be both state variables and functions of those variables. | The C array "yout" may contain more than just the derivatives of the state variables. | The R data.frame "out" contains a row for each time and a column for time plus all variables requested via the "monitor.vars" argument. | The *deSolve* function ode() returns a data.frame with a row for each time requested and a column for each output returned by the C function. |

In the modelinfo_[MODEL].R file the variable "param.names" lists all the parameters used by R, which sometimes includes things such as logP, which underly the parameters used by the ODE system but are not directly needed by the model. The "Rtosolvermap" variable identifies which R parameters link to C parameters described by the "compiled.param.names" variable in the modelinfo file. These, as with States, must be defined in the order used by C and may be used as an additional dosing parameter for particular exposure scenarios.

Outputs describe what the ODE model exports to the R environment. Changes in state variables must be returned. However, it may also include related variables such as concentrations (when state variables are amounts), state variables in different units, mass-balance checks, and time-integrated values (for example, area under the curve - AUC). That is, the C array "yout" may contain more than just the derivatives of the state variables, providing a means of returning values calculated from the state variables to *deSolve* or *httk*.

For further details on the MCSim modeling language and more complex modeling needs we refer the reader to the MCSim documentation (https://www.gnu.org/software/mcsim/mcsim.html) [66].

## Convert MCSim to C

Once we have the MCSim model description file we need to translate the ".model" file to a compilable C script, since R is unable to directly interpret MCSim files. C is a fast machine-specific language [88], which performs rapid model simulation and has an existing package to allow for direct communication between R and C for dynamic modeling (*deSolve* [32]). Generating the compilable C script from the MCSim file requires using the "mod" function from MCSim [66]. The following command executed in a command-line terminal converts an MCSim model file to a C model file:

mod -R < name > .model < name > -raw.c

The generated C version of the model definition file (with extension ".c") will also be accompanied by an initialization file (with extension "_inits.R"). The "_inits.R" file(s) generated during the conversion may be deleted since they are unnecessary for the R package and the functionality is already provided elsewhere in *httk*. Note, we suggest including the suffix "-raw" when generating the C file or to adding it to the initial C file name (model_[MODEL]-raw.c) to denote that the file is an unformatted version of the model definition file.

## Add C version of model to R

R packages that utilize compilable code (such as the.C files) for computational efficiency group these scripts in the "/src" sub-directory. Therefore, to enable the *httk* package to access the C version of your model it must be added to the "httk/src" sub-directory. However, the "raw" C model definition file, as initially generated by MCSim, requires a few minor modifications prior to integration into *httk*.

Most of these modifications include renaming default variable and function names auto-generated by MCSim. MCSim was originally designed to handle one model at a time and assumes the individual models are standalone. Thus, the reuse of default function names in this scenario is not an issue. However, this does not consider the scenario that multiple models may be stored in the same directory as a suite of models as in the *httk* environment. In the *httk* scenario, to avoid confusion amongst the suite of existing models in the package we must rename some of the default function and variables names such that they are unique. Table 4

**Table 4. List of functions and variables auto-generated by MCSim's "mod" function, when converting the model definition file from the MCSim language to C, which must be renamed or removed prior to incorporation into the R package httk.**

| C File Variables and Functions Auto-generated by MCSim's "mod" function. | Purpose | Action Needed for Incorporation into *httk* |
|---|---|---|
| Nout, nr, ytau, yini, lagvalue, CalcDelay | Function definitions for delay differential equations | These variables and functions must be commented out (or deleted). |
| initmod, initforc, getParms | Initializers | These functions must be renamed with the model name suffix in both the.c and the modelinfo_[MODEL] file – for example initmod_[MODEL], intiforc_[MODEL], and getParms_[MODEL] |
| initState | Initializers | This function must be commented out (or deleted). |
| derivs, jac, event, root | Dynamics section | These functions must be renamed with the model-specific suffix, that is derivs_[MODEL], jac_[MODEL], event_[MODEL], and root_[MODEL]. Note that to date no *httk* model has made use of jac, event, or root. However, we anticipate situations where they may be needed for a faster solution to be achieved if they are defined. |

details the parameters and functions that require consideration, in which files they are found, and what updates are necessary.

First, we need to update the standardized names, generated by MCSim to be model-specific. In the newly created "raw" C file, model-specific names for functions and parameters are created by appending the model's name as a suffix. For example, for the Linakis [10] gas inhalation PBTK model the string "gas_pbtk" is the model name suffix that will replace the generally referenced "[MODEL]" suffix found in the function names in Table 4. This designates these functions and parameters as specific to the gas inhalation model in the *httk* model suite. The second modification to the "raw" C file requires removing, or commenting out, unnecessary functions included as part of the MCSim to C code conversion. These include functions or parameters such as "Nout" or "initState", see Table 4 for a full list. Once the necessary modifications are made to the "raw" C file we can save a copy of it, without the "-raw" filename tag, and add it to the "httk/src" sub-directory.

The file "httk/src/init.c" lists descriptions (prototypes) of all the functions *deSolve* needs for all the models in *httk*. Though the file already exists as part of the *httk* package, when a new model is added to the "httk/src" subdirectory this file also needs updated. When adding a new model to the *httk* package, the following function names and argument types must be defined in the "httk/src/init.c" file, for example:

```
extern void getParms_[MODEL](double *, double *, int *);
extern void initmod_[MODEL](void *);

extern void derivs_[MODEL](int *, double *, double *, double *,
  double *, int *);
extern void jac_[MODEL](int *, double *, double *, int *, int
  *, double *, int *, double *, int *);
extern void event_[MODEL](int *, double *, double *);
extern void root_[MODEL](int *, double *, double *, int *, dou-
  ble *, double *, int *);
extern void initforc_[MODEL](void *);
```

In addition, the variable R_CMethodDef must be updated, for example:

```
{"getParms_[MODEL]", (DL_FUNC) &getParms_[MODEL], 3},
{"initmod_[MODEL]", (DL_FUNC) &initmod_[MODEL], 1},
{"derivs_[MODEL]", (DL_FUNC) &derivs_[MODEL], 6},
{"jac_[MODEL]", (DL_FUNC) &jac_[MODEL], 9},
{"event_[MODEL]", (DL_FUNC) &event_[MODEL], 3},
{"root_[MODEL]", (DL_FUNC) &root_[MODEL], 7},
{"initforc_[MODEL]", (DL_FUNC) &initforc_[MODEL], 1},
```

Other than the function names, the types and number of arguments are consistent across models solved with *deSolve*.

## Describe the model within R

After the formatted C model definition file is created and saved in the R package, we then generate a model information file and any necessary parameterization functions (R scripts) based on the defined HTTK model. These R scripts will be stored in the "httk/R" sub-directory of the *httk* package. Each of the R scripts described in this section allow model developers to incorporate their model with the basic functionality of *httk* as well as the more

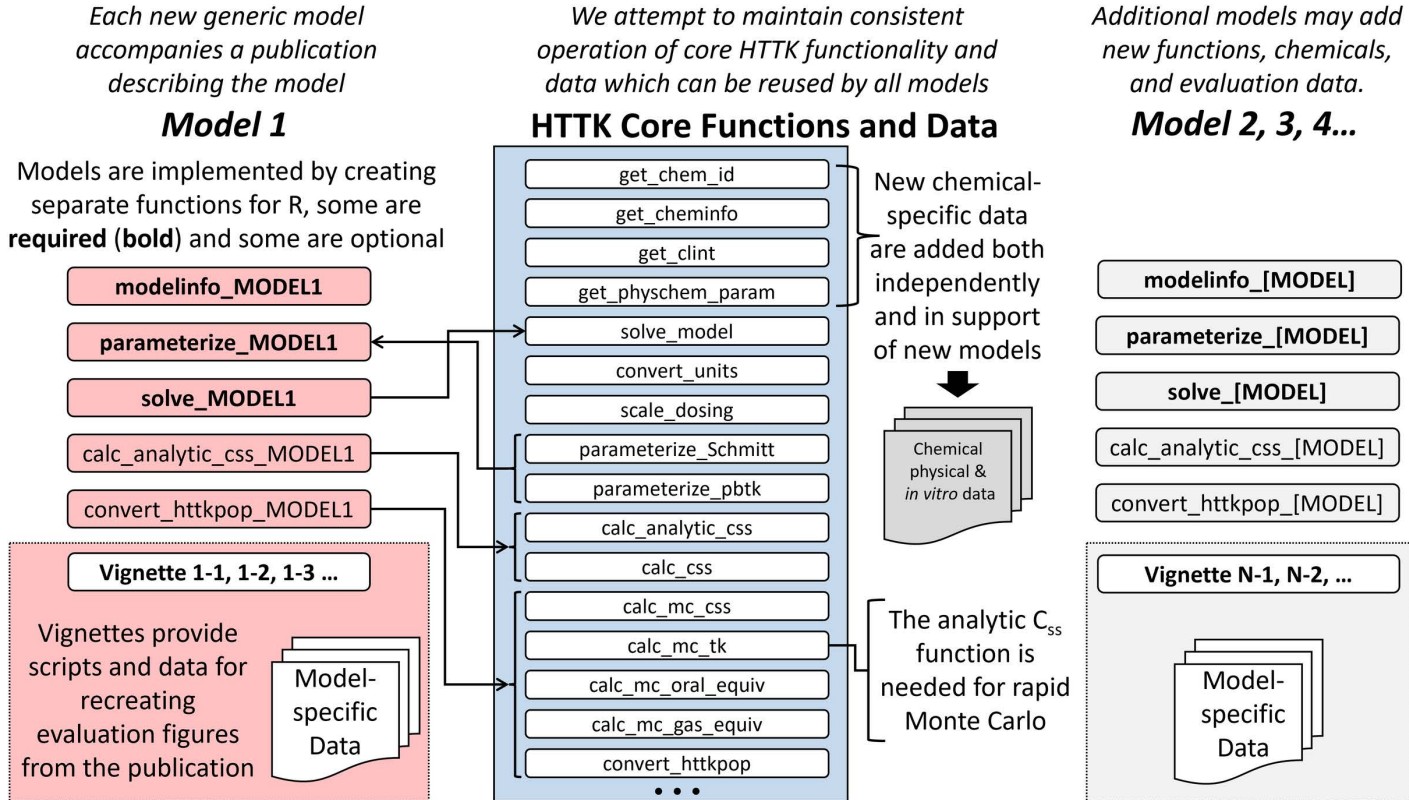

**Fig 4. Schematic of HTTK model specific files, functions, and connections with core *httk* functions. *httk* is intended to permit flexibility with respect to the description of toxicokinetics but allows the reuse of data and functions for tasks such as parameterization, unit conversion, steady state calculation, and Monte Carlo simulation.** Core functions in *httk* are defined in Table 6. We refer the reader to the S2 File for a full list of existing *httk* functions.

complex extensions (for example, steady state modeling or Monte Carlo simulations), while maintaining a standard approach using a set of generalizable 'core' functions provided in *httk*. Fig 4 provides a basic visualization showing the connection between the key auxiliary files – for example the model information, parameterization function, and core functions – enabling the incorporation of the new model into *httk*.

Version 2.0.0 of the *httk* package implemented a set 'core' functions providing a common interface to standard functions such as data reading, parameter entry/modification, and model execution, to generalize interactions with all the models hosted in *httk*. To facilitate these interactions, the model information file (modelinfo_[MODEL].R), more generally referred to as the "modelinfo" file, was developed. The modelinfo file acts as a reference guide for each HTTK model providing model-specific global variables, mappings between R and C variables and functions, as well as criteria indicating data that are relevant for model evaluation. Each modelinfo file saves the standard set of descriptors to the 'model.list' object within *httk*, which holds the model-specific information for all models in the package. The top level of list elements in 'model.list' correspond to a single generic HTTK model allowing it to act as a look-up object when 'core' functions need model-specific information to perform a task. The 'model.list' object allows the 'core' functions to identify and interact with all models in a standardized manner using standardized descriptors.

For a new generic HTTK model, the corresponding modelinfo file needs to create a new list element in the 'model.list' object, namely "model.list[[MODEL]]". Basic model incorporation

requires 1) defining the components that allow communication between the R and C languages, 2) tissue information, 3) applicable data information, and 4) consideration of units. All the basic components are listed in Table 5.

For model developers intending to extend their model with additional *httk* functionalities (for example, steady state calculations, uncertainty propagation, etc.) additional components will need to be included in the modelinfo file. However, we refer readers to the "Generic HTTK model extensions" section for details regarding the various model extension requirements. For additional details demonstrating how to create the modelinfo file we refer the reader to the *httk* vignettes and help files – see S1 File for a full list of possible components.

Once the modelinfo file is generated/updated with the new HTTK model information, the next focus is to create the parameterization function script (parameterize_[MODEL].R). The function in this script interacts with a larger set of functions and data in *httk* defining a series of steps to obtain and format *in vitro*, physiology, and physical-chemical property (physico-chemical) data necessary for parameterizing and solving the new HTTK model. In some cases, there is no dynamic modeling component (that is, no C script) and the parameterization function is the only necessary component to run the model (for example, the parameterize_schmitt() function). Regardless of the type of parameterization function being developed the essential information typically includes – but is not limited to – relevant physiology data, chemical identifiers, relevant tissues, and physico-chemical parameters.

While some models use their own approach for retrieving chemical-specific model parameters and generating model parameterization data, many are developed around the existing core function parameterize_pbtk(). The parameterize_pbtk() function can be customized to create a model-specific function by specifying which tissues are lumped together into aggregate compartments. For example, the parameterization function for the 3-compartment model, parameterize_3comp(), makes use of parameterize_pbtk() to obtain all necessary parameters and sets any unnecessary parameters to NULL. More complex models may use parameterize_pbtk() several times in their model-specific parameterization function. As mentioned previously in the results, one such model is the maternal-fetal model from Kapraun [11], with the corresponding parameterize_fetal_pbtk() function, which calls

**Table 5. Basic 'model.list' elements to include in the modelinfo file when incorporating a new generic HTTK model into httk. Note, this table only covers the basic model and data criteria components not any model extension components. We refer readers to the S1 File for further details on all components.**

| Component Type | Model Specific List Components | List Component Functionality |
|---|---|---|
| Basic/ Dynamic | parameterize.func, solve.func | R Functions |
| | alltissues, tissuelist | Tissue Information |
| | param.names, required.params | R Parameters |
| | Rtosolvermap | R to C Parameter Mapping |
| | compiled.param.names, derivative.output.names | C Parameters |
| | compiled.parameters.init, compiled.init.func, derivative.func | C Functions |
| | allowed.units.input, allowed.units.output, compartment.units | Default Unit Information |
| | default.monitor.vars, state.vars | C Variables |
| | routes | Dosing Components |
| | forcings.materials, input.var.names | Forcing Dose Components |
| | compartment.state | HTTK Rout and Chemical State Components |
| Data | exclude.fup.zero, log.henry.threshold, chem.class.filt | Data Inclusion/Exclusion Criteria Components |

the parameterize_pbtk() function twice. The first call obtains parameters for the mother and the second call obtains parameters for the fetus. However, if a modeler needs a more flexible parameterization approach, they may consider creating a new one from scratch and utilize other core functions that accomplish similar tasks – for example get_chem_id(), get_clint(), get_physchem_param(). For any new modeling scenario, a modeler may need to include core functions and data as part of the parameterization function which are discussed in more detail in the "General utility core function" section. We refer readers to the *httk* vignettes for further details on creating a parameterization function for a new HTTK model, which may have a unique set of needs. Once the modelinfo and parameterize function R scripts are created they should be saved in the "httk/R" sub-directory of the *httk* R package.

## Using core functionality

In addition to the modelinfo and parameterization function R scripts, developers may want to automate running their new generic model. Most HTTK models have a "wrapper" function around the generic solve_model() function (that is, solve_[MODEL].R). The solve_[MODEL]() wrapper function calls solve_model() function internally and adds additional tailoring arguments specific to the new model, for example solve_gas_pbtk(). "Wrapper" functions generally configure, evaluate, and/or extend basic functionalities of the primary analysis by encapsulating it with recurrent steps used before or after the primary analysis/function call – for example, data retrieval and formatting. Some recurrent steps can be accomplished using core *httk* functions, such as pulling physico-chemical data or performing unit conversions. Generating a wrapper function allows the model prediction process to be generalizable to the extent possible given model constraints and applicability. It enables evaluation across a variety of scenarios, standardization of the process, and potential for mitigating errors. Though creating a wrapper function is not required for specific HTTK models, it does have two major advantages. First, it allows for more reproducible results, depending on the function configuration. Second, it encourages more wide-spread application of the new model by making it readily available for *httk* users that are not necessarily modelers but want to make use of the model's predictions for their own analysis.

As mentioned previously, solve_model() is a generic function, that is usable for any HTTK model, used to communicate with the compilable C code for an HTTK model via the R package *deSolve* [32]. *deSolve* interfaces between R and C to obtain a solution to the system of ODEs. Thus, solve_model() is designed to accept systematized data and metadata for a given toxicokinetic model, including – but is not limited to – variable names, parameterization functions, and key units. When this model description information is passed to solve_model(), along with chemical-specific descriptors, the function can set up the ODE model system to obtain the numerical solution of chemical amounts (or concentrations) in different body compartments over time. Much of the information solve_model() requires is either specified in the modelinfo file – described previously – or pulled from *httk*-provided datasets using core functions like get_cheminfo() or get_physchem().

It is imperative that the data provided to solve_model(), and passed to the C model, are in the units anticipated by the model. Thus, the convert_units() and scale_dosing() functions are commonly employed within solve_[MODEL]() wrapper functions. This allows users to provide their experimental data "as-is", that is without any unit conversion, to the wrapper function and the unit conversion and/or scaling functions employed within ensure the data are appropriately converted to the expected units in a seamless and repeatable manner. Once the solve_[MODEL]() function R scripts is created it should be saved in the "httk/R" sub-directory of the *httk* R package. For further details on constructing a solve_[MODEL]()

wrapper function, we refer users to the *httk* vignette demonstrating how to create and document the function along with some considerations when developing a new HTTK model.

## Installing a new version of *httk*

Once all the necessary files for the new HTTK model are generated and saved in their respective sub-directories we need to compile and install the package prior to execution. It will likely be necessary for Windows users to install rtools: https://cran.r-project.org/bin/windows/Rtools/rtools44/rtools.html. One can compile and install the *httk* package using the following two commands executed in a command-line terminal:

```
R CMD build httk
```

```
R CMD INSTALL httk
```

Alternatively, one can use the build() and install() functions, from the R package *devtools* [89], in the R console to compile and install the new *httk* package. We recommend changing the version number in the DESCRIPTION file when a new model is added to reduce confusion. Typically, we denote unreleased development versions of *httk* with the major number (first digit) "99" – for example, "version 99.3.10".

Re-installing the package enables the new model to utilize existing functionalities, package dependencies, and data to solve the system of ODEs in the new generic HTTK model. Once the updated *httk* package is installed with the newly incorporated model it is possible to obtain concentration vs. time predictions with that new generic HTTK model.

## General utility core functions

Throughout the past few sections, we gave a high-level description of the process for compiling and incorporating a new generic HTTK model into the *httk* R package. In this section, we highlight some of the core functions in *httk* that enable modularity and harmonization of models across the package. Core functions are generalized, yet 'model-aware', and can be used with any generic HTTK model added to *httk* (see Fig 4). The model-aware attributes of core functions allow for tailored results based on the model with which they interact.

Consider, for example, that each generic HTTK model requires different parameters. Therefore, the chemicals with sufficient descriptive parameters may vary model-to-model. In this instance, the get_cheminfo() function serves to identify and report chemicals applicable for the specified model. For example, if your model requires fraction unbound and intrinsic clearance, get_cheminfo() will only identify the chemicals that have those parameters available. Optional arguments in the core functions give users the flexibility to tailor results for their needs. Continuing with our example function, get_cheminfo(), the "info" argument defaults to returning only the Chemical Abstract Services Register Number (CASRN) identifiers, but a user may also specify other data to include. To include additional information, users provide a character vector specifying additional data to retrieve – for example, logP and molecular weight. Another optional argument necessary for some models is "class.exclude", which checks the model information file to see if there are any chemical classes that are known to be outside the domain of applicability for the specified model and ought to be excluded. Though we will not provide an exhaustive list here of all available core functions in *httk,* and their arguments, Table 6 outlines several commonly used functions with a brief description of their utility. For additional details we point the reader to the *httk* help files and the S2 File which have exhaustive lists of available functions.

**Table 6. Reusable core functions with general utility for generic TK models. (Note, this table only lists a subset of core functions available in the httk R package and is limited to functions explicitly mentioned in this paper. We refer readers to the S2 File and help files for further details on these and other available functions.).**

| HTTK Core Function | Description |
|---|---|
| get_chem_id() | Interface for users to check if a chemical of interest is available within *httk* data and returns all three possible chemical identifiers (that is, chemical name, Chemical Abstract Services Register Number (CASRN), and DSSTox substance identifier (DTXSID)). |
| get_clint() | Interface for users to retrieve chemical- and species-specific intrinsic hepatic clearance ($Cl_{int}$) for a given list of chemicals. |
| get_physchem_param() | Interface for users to retrieve physico-chemical properties from the chem.physical_and_invitro.data table in *httk* for a given list of chemicals. |
| get_invitroPK_param() | Interface for users to retrieve *in vitro* pharmacokinetic (PK) data, for example intrinsic metabolism clearance or fraction unbound in plasma for a given list of chemicals and species. |
| get_cheminfo() | Interface for users to list the chemicals with sufficient data to run a specified HTTK model. Interacts with internal HTTK data tables containing chemical-specific information. [NOTE: By default, the function assumes the model is "3compartmentss" and reports the chemical identifiers as CASRN.] |
| add_chemtable() | Utility function that adds chemical-specific information to the chem.physical_and_invitro.data table necessary to parameterize a model, such that additional chemicals can be modeled/evaluated. |
| solve_model() | Generic numerical ODE solving function that uses *deSolve* to call the C code for the derivative and simulate a concentration vs. time response. Often used by wrapper functions (for example, solve_gas_pbtk(), solve_steadystate(), and solve_pbtk()). |
| convert_units() | Utility function for converting input and output values from their current units to the appropriate units necessary for model or user interpretation, respectively. |
| scale_dosing() | Utility function converting dose in terms of unit per body weight to specified dose units not in terms of body weight. |
| parameterize_schmitt() | Utility function providing the necessary parameters to estimate predictions for plasma partition coefficients [34,35,90]. |
| parameterize_pbtk() | Utility function generating a set of chemical- and species-specific PBTK model parameters, including tissue:plasma partition coefficients, organ volumes, and flows for lumping tissues. Tissues should be described in the tissue.data data.frame. |

The two most crucial core functions mentioned in Table 6, which we highly encourage model developers to use when incorporating a new model, are the convert_units() and scale_dosing() functions. These functions are crucial because they implement tools that adhere to the "Rules of PBTK Modeling" (commonly featured in classes on PBTK modeling by Dr.s Mel Andersen and Harvey Clewell, in addition to others from Wright-Patterson Air Force Base in the United States):

(1) Check Your Units

(2) Check Your Units

(3) Check Mass Balance

Unit conversions are among the most common sources of PBTK model errors [45]. Thus, the convert_units() and scale_dosing() functions aim to mitigate this source of error by defining the relevant calculations in one function and calling the relevant function when a conversion is necessary. The scale_dosing() function performs species-specific body weight scaling by detecting "/kg" in unit strings, which indicates units in per kilogram body weight, then scales the dose values with the respective multiplier (body weight). Alternatively, the convert_units() function provides conversions for many other units, specifically amounts and concentrations (that is, amount per unit volume). For example, convert_units() can be used to convert "mg" to "μmol", or "mg/L" to "μM". It should be noted, for conversions between concentrations and amounts – for example "mg" to mg/L" – a volume value is required. Implementing reusable unit conversion functions minimizes the burden on users to manually include these calculations in their code and avoids the unintentional introduction of bugs due to typographical errors. More importantly, it centralizes the code where these conversions are

defined. This greatly facilitates evaluation of the units used in inputs, in calculations and in producing results such as figures. We refer readers to *httk* help files and vignettes for further details on using these functions, how to add new conversions, and other details to consider.

### Generic HTTK model extensions

Users may want to go beyond performing *in vitro* to *in vivo* extrapolation (IVIVE) for various exposure scenarios and quantify levels of uncertainty and variability in data and assess their impacts on model predictions. Particularly for Monte Carlo simulation, users may want to calculate tissue concentrations when steady state is reached (that is, metabolic equilibrium). In the *httk* package, we extend the basic functionality of HTTK models with generic functions for performing steady state calculations and uncertainty estimation via Monte Carlo simulations. Table 7 provides a list of reusable generic functions for extending the functionality of a new model.

### Steady-state modeling

For chemical exposure simulation we are rarely interested in true "steady state" (wherein none of the state variables of the model are changing). However, we are often very interested in quasi-steady state concentrations, wherein after a sufficient time has passed for a relevant repeated discrete dosing scenario (for example, dosing three times a day). In this case, "steady state" concentrations eventually oscillate around some average value. If the generic HTTK model in development is appropriate for steady state modeling, then incorporating this functionality is much like the other auxiliary functions for the model described previously.

The first thing that needs to be created is the model-specific analytic steady state function, namely calc_analytic_css_[MODEL]. All the necessary model equations and calculations for estimating the model-specific steady state values should be included in the calc_analytic_css_ [MODEL] R function. It should be noted that the model-specific steady state function created for a model will not be used directly. Rather, it will be used via a generic wrapper function, similar to using the solve_[MODEL]() function to interact with a specific HTTK model which then calls the solve_model() function to communicate directly with the C code for obtaining a model solution. Here, calc_analytic_css() and calc_css() are the generic wrapper functions that access the model-specific steady state estimation function calc_analytic_css_[MODEL]().

**Table 7. Reusable functions for extending the functionality of generic HTTK models. We refer readers to the S2 File and help files for further details on these functions and other existing functions.**

| Generic Functions | Basic HTTK Extension | Description |
|---|---|---|
| calc_analytic_css() | Steady state Modeling | Calculates the quasi-steady state plasma concentration by calling the model-specific analytical solution when it is available for the specified model. |
| calc_css() | Steady state Modeling | This is a numerical approach to determine the quasi-steady state concentration (Css) as a function of repeated dosing. Dosing can be changed to match different conditions (default is three times a day). Also, returns the days needed to reach steady state. |
| calc_mc_tk() | Monte Carlo Simulation | Function for running a Monte Carlo simulation with a TK model. Essentially runs solve_model() for a range of parameter sets. |
| calc_mc_css() | Monte Carlo Simulation | Function for calculating Css (quasi-steady state plasma concentration) using Monte Carlo simulation of Css – typically uses the analytic solution since it is much faster than the numerical approach. |
| calc_mc_oral_equiv() | Monte Carlo Simulation | This is the "reverse dosimetry" IVIVE function that uses the Css quantiles from calc_mc_css() to estimate a daily oral equivalent dose based on quasi-steady state plasma concentration using Monte Carlo simulation. |
| calc_mc_gas_equiv() | Monte Carlo Simulation | This is a new "reverse dosimetry" IVIVE function from Breen (in preparation) for daily inhalation rate that would induce a given plasma concentration at steady state. |

**Table 8. Model information file components necessary to use steady state modeling. We refer readers to the S1 File for further details on all components.**

| Model Information File Component | Description of component |
|---|---|
| analytic.css.func | Character string with the name of the model-specific steady state function, that is "calc_analytic_css_[MODEL]". |
| steady.state.compartment | Character string for the model compartment whose steady state concentration is modeled. Typically, only one compartment. |
| steady.state.units | Character string providing units for the steady state modeling results returned. |
| css.dosing.param | Vector of character strings naming the parameters necessary to specify dosing protocols for model-specific steady state modeling, for example "hourly.dose". |

To enable access, the corresponding generic HTTK model needs several components to be defined in the model information file – including analytic.css.func, steady.state.compartment, steady.state.units, and css.dosing.params. Table 8 provides a brief description for each of the necessary model information components.

After the steady-state function R script (calc_analytic_css_[MODEL].R) is added to the "httk/R" sub-directory and the components for the generic HTTK model of interest are included in the modelinfo file, the calc_analytic_css() and calc_css() functions can perform steady state modeling when the argument "model" indicates the generic HTTK model of interest. For further details on constructing the calc_analytic_css_[MODEL] function and updating the model information file, we refer the reader to the help files and vignette in the *httk* package.

## Monte Carlo simulation

Within *httk* uncertainty and variability in the model parameters can be propagated to the model output with Monte Carlo (MC) simulation [43]. In MC simulations, model parameters are sampled from user-specified distributions and the model is evaluated for each set of parameters. Parameters may be sampled from the same or different distributions depending on the user's preferences. For physiological parameters in particular, the user may choose to apply *httk*'s built-in population physiology sampler known as "HTTK-Pop" [82]. Here, we describe the updates necessary for both types of MC simulation together and denote which updates are specific to HTTK-Pop. It should be noted that use of HTTK-Pop is not required, and it should not be used for non-human populations. Extending the basic functionality of a generic HTTK model to include MC simulation capability follows a process similar to the steady state modeling extension described above.

To understand the necessary components for enabling MC simulations with a new model, it is useful to understand *httk*'s built-in wrapper functions that perform MC simulations. The built-in wrapper functions for MC simulation, such as calc_mc_css() and calc_mc_oral_equiv(), were described previously in Table 7. Both calc_mc_css() and calc_mc_oral_equiv() require the analytical steady state calculation function, calc_analytic_css(), already be incorporated – see the "Steady-state modeling" section for further details. Fig 5 provides an overall schematic of the MC simulation process, and highlights some of the model-specific pieces needed when extending an HTTK model.

Both wrapper functions, use create_mc_samples() to generate a table of MC-sampled model parameters. Users may also call create_mc_samples() directly. To incorporate a new model into the existing built-in MC wrapper, functions require the user to define several components in the model information file. Table 9 provides a list of MC simulation components

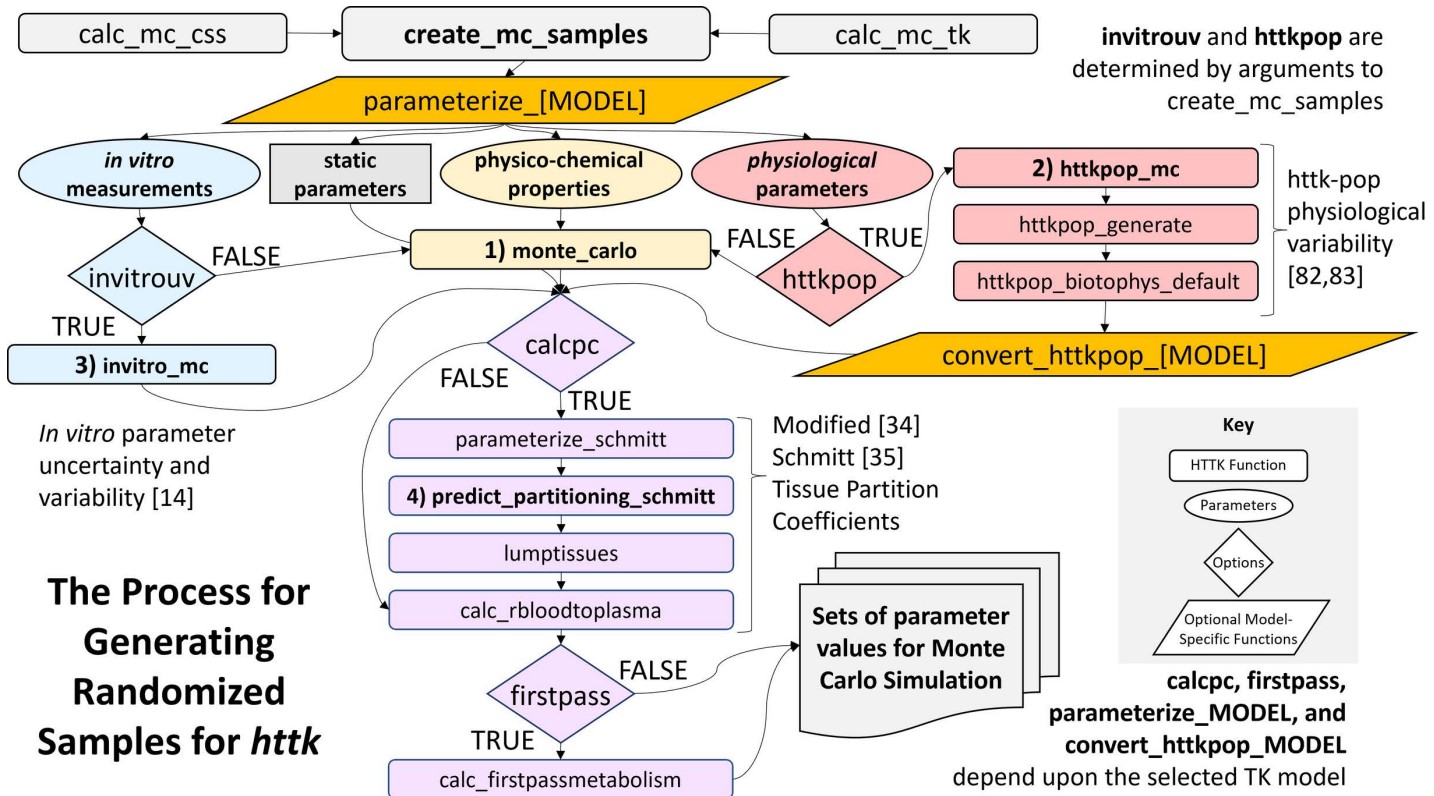

**Fig 5. Schematic of the overall Monte Carlo (MC) simulation process in *httk*.** This schematic outlines how uncertainty and variability is propagating via MC simulation in *httk* (figure adapted from Breen [83]). The blue box, create_mc_samples(), denotes the built-in wrapper function coordinating model-specific information from the modelinfo file and user specified arguments to generate MC samples. The yellow boxes, parameterize_[MODEL]() and convert_httkpop_[MODEL](), are model-specific functions that may be necessary for integrating MC sampling. The red boxes, "calcpc" and "firstpass", denote model-specific logical arguments specified in the modelinfo file for a given HTTK model.

necessary for the model information file with a brief description of each. It should be noted that some components are HTTK-Pop specific, and some are optional based on additional functionality that may be relevant for a given model.

The create_mc_sample() function uses the components listed in Table 9 at various points in Fig 5. Here, we will briefly describe how create_mc_sample() works and explain when each component is required. First, create_mc_samples() generates the default set of parameters for the model, using the parameterize_[MODEL]() function defined under "parameterize.func", see the "Creating and integrating a new HTTK model" section for further details. Next, MC simulation of parameters is done in four steps, some of which may be optional depending upon relevant functionality. These steps include drawing samples for relevant parameters from truncated/censored distributions, sampling physiological parameters using HTTK-Pop, sampling chemical-specific *in vitro* parameters, and recalculating parameter values that depend upon other parameters.

Step One draws samples for any parameters listed in the "censored.params" and/or "vary.params" arguments of create_mc_sample() from censored and/or truncated normal distributions, respectively. Nothing needs to be defined in the modelinfo file to complete this step. However, if the components "httkpop.params" or "invitro.params" are defined (see Fig 5), they are used to check whether a parameter sampled as part of this step will be overwritten

**Table 9. Model information file components necessary to use Monte Carlo simulation. This list does not include the elements parameterize.func and param.names, which are assumed to be part of the existing basic components in the model information file – see Table 5. We refer readers to the S1 File for further details on all components.**

| Model Information File Component | Description of component |
|---|---|
| calcpc | Logical variable indicating whether partition coefficients should be recalculated for each simulated individual during the MC simulation process. |
| firstpass | Logical variable indicating whether first pass metabolism should be recalculated for each simulated individual during the MC simulation process. |
| calc.standard.httkpop2httk | (HTTK-Pop Only) Logical variable indicating whether HTTK-pop generated parameters should be converted to standardized HTTK physiological parameters using httkpop_biotophys_default(). |
| httkpop.params | (HTTK-Pop Only) Vector of character strings naming the model parameters to be sampled using HTTK-Pop. Should be a subset of param.names. |
| propagateuv.func | Optional: Character string with the name of the model-specific function for additional uncertainty and variability propagation to necessary model parameters, that is "propagate_invitrouv_[MODEL]". |
| invitro.params | Optional: A vector of character strings naming the model parameters to be varied using invitro_mc(). Used only to check whether any parameters sampled using invitro_mc() are also specified to be sampled from a truncated/censored normal distribution and produce a message that warns the user accordingly. |
| convert.httkpop.func | (HTTK-Pop Only) Optional: Character string with the name of the model-specific function for converting HTTK-pop physiology into relevant model parameters, that is "convert_httkpop_[MODEL]". The first argument of this function must be "parameters.dt" (a data.Table [91] of model parameters with names corresponding to param.names), and the second argument of this function must be "physiology.dt" (a data.table of physiological parameters as returned by httkpop_mc()). If convert.httkpop.func is NULL (that is, not assigned any value), then the output of httkpop_mc() will be used as-is. |

in Steps Two or Three of MC simulation and will produce a warning message for the user, if necessary.

Step Two, HTTK-Pop sampling, only occurs if the "httkpop" and "species" arguments for create_mc_samples() are set to TRUE and "human", respectively. The httkpop_mc() function will sample physiological parameters from distributions based on the United States population as surveyed by the United States Centers for Disease Control and Prevention National Health and Nutrition Examination Survey [83,92,93]. If a modeler desires parameters returned by httkpop_generate() to be converted to standardized units expected by the existing HTTK models (that is, directing httkpop_mc() to run httkpop_biotophys_default()), then the "calc.standard.httkpop2httk" in the modelinfo file should be TRUE. For further details on the httkpop_biotophys_default() function we refer readers to the *httk* help file. The resulting sampled parameter table will be returned to create_mc_samples(). When additional conversions or calculations on the parameters output by httkpop_mc() are necessary, the user may optionally define a model-specific function, namely convert_httkpop_[MODEL](), and provide the name to the "convert.httkpop.func" component in the modelinfo file. If "convert.httkpop.func" is defined, then create_mc_samples() calls the specified function. Otherwise, the output of httkpop_mc() will be used as-is. After httkpop_mc() and convert_httkpop_[MODEL]() (if used) are called, parameters in the resulting table will be retained only if they appear in the parameter list returned by the parameterize_[MODEL]() function.

Step Three, if the "invitrouv" argument for create_mc_samples() is TRUE, then the invitro_mc() function performs Monte Carlo sampling for *in vitro* chemical-specific parameters

– such as intrinsic hepatic clearance ($Cl_{int}$), fraction unbound in plasma ($f_{up}$), and Caco2 membrane permeability. Nothing needs to be specified in the modelinfo file for this step.

Step Four, the create_mc_samples() function recalculates parameter values that depend on the values of other parameters – particularly values of *in vitro* chemical-specific parameters sampled in Step Three. Modelers may control this process by specifying the "calcpc" and "firstpass" components, both of which are logical – that is TRUE/FALSE, in the modelinfo file. If "calcpc" is TRUE, then the tissue:plasma partition coefficients are recalculated for each simulated individual using the individual's value for fraction unbound in plasma (which may be varied in Step Three) and other relevant parameters. If "firstpass" is TRUE, then the partition coefficients are recalculated, and first-pass hepatic bioavailability is also recalculated for each simulated individual. For additional model-specific uncertainty/variability propagation that may be necessary, a modeler may define a model-specific function, namely propagate_invitrouv_[MODEL](). For example, propagate_invitrouv_1compartment() computes "Vdist" (volume of distribution) and "kelim" (elimination rate), in addition to the standard parameters, using the other sampled parameter values for the one compartment model. The first argument for the propagate_invitrouv_[MODEL]() function should be "parameters.dt", which accepts a data.table of model parameters. Other arguments available for the function are those that can be passed into the "propagate.invitrouv.arg.list" argument of create_mc_samples(). A data.table of model parameters is the expected output for this function. If a modeler defines a model-specific uncertainty/variability propagation function, then it should be specified in the modelinfo file component "propagateuv.func".

After the creating the model-specific functions and updating the relevant components in the model information file, for the generic HTTK model of interest, the create_mc_samples() function can generate the necessary samples for MC simulations with the argument "model" indicating the model of interest. We refer the reader to the *httk* help files and vignettes for further details on constructing functions and updating the model information files for MC simulations.

## Data library for HTTK models

In addition to the core functions, *httk* also provides a set of data tables with information needed to parameterize HTTK models. These tables help enable modularity and consistency by eliminating the need to hard code model parameter values. There are several key datasets used to parameterize and evaluate generic HTTK models. These datasets are stored in the Tables.RData file under the "httk/data" sub-directory. Table 10 provides a list of core datasets that are automatically loaded into a local R session when *httk* is called via "library(httk)". These include, but are not limited to, two chemical and two physiological parameter tables as

**Table 10. Core HTTK data sets – includes chemical specific tables with physico-chemical properties, physiological data, and data generation meta-data.**

| Data Object | Description |
| --- | --- |
| chem.physical_and_invitro.data | Chemical-specific values for physico-chemical properties and *in vitro* measurements. |
| chem.invivo.PK.data | Chemical concentration vs. time database (CvTdb) snapshot for evaluating model predictions [22]. |
| physiology.data | Data describing species-specific physiology. |
| tissue.data | Tissue composition data for Schmitt's method of partition coefficient prediction. |
| Tables.Rdata.stamp | Time stamp identifying when the tables were created. |

well as a metadata object with information about the date of the data table generation. Users can retrieve parameter values stored in these tables with the corresponding core functions. Though there are other tables available in the package, aside from those listed in Table 10, they are not considered part of the core set necessary for parameterizing and solving HTTK models. Here, we describe the four core datasets and their role in incorporating a new model in *httk*. We refer the reader to the *httk* vignettes and help files for details about using these and all other available datasets.

## Chemical specific data

As of 2024, 1046 chemicals had *in vitro* measured values collected from the literature and made available with *httk*. Each of these chemicals has both *in vitro* measured plasma protein binding and intrinsic hepatic clearance that can be used with generic TK models. Some, simpler TK models can be used with just intrinsic clearance and the assumption that plasma protein binding measurements were attempted but failed because the chemical was highly bound (a default fraction of 0.5% unbound is assumed) [4,9]. Various structure-based models have been used to provide additional, overlapping sets of *in silico* predictions for 8719 [19], 8573 [94], and 10,452 [95] chemicals. These *in silico* predictions are available via the commands load_sipes2017(), load_pradeep2020(), and load_dawson2021() functions, respectively.

The chemical-specific *in vivo* and *in vitro* measurements as well as the physico-chemical properties in the chemical.physical_and_invitro.data dataset make it the most used parameter table. The chemical.physical_and_invitro.data dataset contains many of the parameters vital for estimating absorption, distribution, metabolism, and elimination of a chemical from the body after an exposure – for example molecular weight or hydrophobicity (that is, logP). There are four key types of chemical-specific parameters available for use with HTTK models – including hepatic clearance ($Cl_{int}$), fraction unbound in plasma ($f_{up}$), blood-to-plasma ratio (Rblood2plasma), and membrane permeability (oral absorption as characterized by Caco-2). More detail on these parameters is provided at the end of this section.

The chemical-specific *in vitro* measurements are based on previously published experiments using human primary cells. In the previously published studies, the vendors providing the cells had obtained relevant consent. ThermoFisher states that "Yes, we comply with country-specific legal and ethical standards for procurement of human liver tissue, including the global ICH Guidelines, and the US's HIPAA, Uniform Anatomical Gift Act, National Organ Transplant Act, and Hospital Internal Review Board (IRB) approval processes." The names of the IRB's used by the vendors in the literature studies are not available.

The chemical-specific *in vivo data* measurements are largely from animal experiments that have been curated from the literature [22]. However, human-specific physiological data [83] are used from the U.S. Centers for Disease Control National Health and Nutrition Examination Survey which is overseen by the National Center for Health Statistics Ethics Review Board. The most survey data are covered by Protocol @2021-5.

Though all the *httk* functions retrieve chemical-specific data from chem.physical_and_invitro.data, few of them directly interact with that data table. There is a cascade of decisions for selecting the appropriate value from chem.physical_and_invitro.data. Decisions include identifying the appropriate species-specific value (or surrogate) and whether *in vivo* or *in vitro*-derived values are to be used. Several functions provide an intermediate layer interacting with chem.physical_and_invitro.data, including: get_cheminfo(), get_invitroPK_param(), get_physchem_param(), and add_chemtable(). The first three functions are useful when a new model needs to pull chemical-specific data as it does this consistently. The fourth function, add_chemtable(), allows a modeler to insert additional data that may be needed to evaluate the new model, particularly adding data on new chemicals. Adding new descriptors (such as

physico-chemical properties) with add_chemtable() currently requires modifications of the function. We refer the reader to the package help files for further details about using these functions and core dataset when developing and evaluating their new model.

Most of the *in vitro* data are measured from human tissues. However, for certain calculations a user may want to use values from non-human physiologies –for example, rat, mouse, etc. – to solve a model. The "default.to.human" argument is provided for many of the parameterization functions to allow this option. In cases where the human values are preferred in lieu of non-human species, users should set "default.to.human" to TRUE.

Intrinsic hepatic clearance ($Cl_{int}$), for the purposes of *httk*, is an *in vitro* measurement for the rate of disappearance for a compound when it is incubated with hepatocytes [96]. $Cl_{int}$ is interpreted as a measure of metabolic rate by the liver, which for many chemicals is the primary organ of metabolism. In *httk,* $Cl_{int}$ is stored in units of $\mu$L/min/$10^6$ hepatocytes. When scaled appropriately, $Cl_{int}$ is predictive for the clearance rate of a compound from the body via the liver. $Cl_{int}$ adjustments are executed via the calc_help_fu() function in *httk*, which accounts for free fraction of chemical in the hepatocyte assay [97]. The calc_hep_clearance() function uses $Cl_{int}$ values to estimate whole-liver hepatic clearance based on various scaling models described by Ito and Houston [98], typically assuming the well-stirred model when the liver is not modeled as a separate compartment. The $Cl_{int}$ parameter may be reported as a single value or a 4-tuple string of values separated by commas. For 4-tuple string of values, these are Bayesian estimates where the median is first, followed by the lower and upper 95th percentile values, and concluded with the p-value for systematic clearance (that is, the likelihood that there was no clearance observed), respectively. If users extend their model with the Monte Carlo sampler, discussed in the "Monte Carlo simulation" section, this information will be used, when available, to appropriately propagate chemical-specific uncertainty throughout the model.

Fraction of chemical unbound in plasma ($f_{up}$) is a central parameter for estimating other predictors within a generic HTTK model. There are data from two different *in vitro* assays measuring $f_{up}$ included in *httk*: rapid equilibrium dialysis [99] and ultracentrifugation [100]. *httk* assumes $f_{up}$ is constant across time and throughout the body. The $f_{up}$ parameter may be reported as a single value or a 3-tuple string of values separated by commas. For 3-tuple string of values, these are Bayesian estimates of $f_{up}$ where the median is first, followed by the lower and upper 95th percentile values, respectively. Finally, $f_{up}$ may require adjustments to account for differences between available lipid amounts *in vitro* vs *in vivo* [34]. These adjustments can be made using the calc_fup_correction() function.

Oral absorption, also known as Caco2 membrane permeability, is the last chemical-specific parameter necessary for many of the *httk* models. This parameter is used for estimating the amount of a chemical taken up in the body via ingestion (oral exposure). Honda [101] describes *in vitro* measured Caco-2 permeability data and a quantitative structure-property relationship model for estimating chemical absorption. The function get_fabsgut() will first check for any existing *in vivo* data on bioavailability, and if not present will use calc_fbio.oral() to calculate fraction absorbed based on Caco-2 permeability. The calc_fgut.oral() function is available to calculate the fraction surviving a rate of gut metabolism estimated from the hepatic clearance as 1% $Cl_{int,hepatic}$. If needed, depending on how flows are described in the model, calc_hep_bioavailability() can also be used to calculate the fraction surviving first-pass hepatic metabolism.

## Calculated chemical parameters

Chemical-specific tissue-to-free-fraction-in-plasma partition coefficients in *httk* describe the partitioning of a chemical into various tissues (that is, equilibrium tissue:free plasma concentration ratios). These partition coefficients are estimated using a calibrated variant of the Schmitt method [35]. The specific implementation within *httk* is described in more detail by

Pearce [34]. New tissues may be specified if certain composition descriptors – for example, fraction lipid – are provided. Any model in *httk* may predict partition coefficients via the function predict_partitioning_schmitt(), which predicts the partition coefficient for each tissue in the tissue.data table. Furthermore, users may "lump" (that is, aggregate) multiple tissue types into a single compartment with a common chemical concentration using the function lumptissues(). The specific lumping scheme (if any) used by a model is described in the corresponding model information file.

Blood to plasma ratio is another necessary parameter since most of the PBTK models in *httk* describe flows in terms of blood but concentrations in terms of tissue to plasma partition. The blood to plasma ratio helps scale available chemical fraction estimates for processes throughout the body such as partitioning, metabolism, and glomerular filtration. The available_rblood2plasma() function works through three options to obtain the most accurate blood to plasma concentration ratio. First, the function tries to find a species-specific measured value, otherwise it defaults to human measured values. Second, available_rblood2plasma() uses Schmitt's method to estimate the red blood cell to plasma partition coefficient along with the calc_rblood2plasma() function. Finally, when insufficient chemical-specific physico-chemical data exists for the first two options the average measured value is used.

## Adding new data

Existing parameterization tables are updated to incorporate new data as it becomes available. Thus, the Tables.RData file, described previously, evolves over time and is potentially unique for each version of *httk*. The "Tables.Rdata.stamp" object tracks when the *httk* data tables were generated (see Table 10). The "datatables" sub-directory in the Git repository contains both the source data files and the R script "load_package_data_tables.R", which is used to generate the Tables.RData file from the source/"raw" data. If there is a need for a user to include additional data, one can update the load_package_data_tables.R script to re-generate the Tables.RData file with the new data. For further details, we refer the reader to the GitHub repository for *httk* (https://github.com/USEPA/CompTox-ExpoCast-httk/tree/main/datatables). To update the available data in *httk*, the re-generated Tables.RData file must be added to the "httk/data" sub-directory.

New physico-chemical properties can be obtained from a number of sources, but *httk* currently retrieves values predicted by OPERA [102] from the CompTox Chemicals Dashboard [51]. We use the R package "ctxR" to access predicted properties through a public Application Programming Interface [75].

Whenever new data is added, or datasets are updated, the data documentation should also be updated. The "httk/R/data.R" file in *httk* contains documentation for all data in the package. Each dataset description should specify the data object name, column names, variable descriptions, and relevant measurement units – further details on documenting R package data can be found in Section 7.1.2 of (https://r-pkgs.org/data.html) [85]. Once the documentation is complete, functions in the R package *roxygen2* [69,103] can be used to generate or update the help files accompanying data objects in *httk*.

## Tools for model evaluation

**Data for model evaluation.** Model evaluation is a key part of determining whether a new HTTK model is ready for inclusion into *httk* and appropriate for use. To facilitate model evaluation several datasets are included in *httk* to aid in model evaluation. A table of

concentration vs. time data is included in *httk* as a flat file along with empirical TK parameter estimates from CvTdb [22]. The chem.invivo.PK.data table, mentioned in Table 10,

contains the concentration vs. time data for several dozen chemicals across multiple species (chiefly rat), often with multiple dose regimens (such as oral vs. intravenous and doses). The empirical TK estimates are included in the chem.invivo.PK.summary.data and chem.invivo.PK.aggregate.data tables, which respectively contain the estimated TK statistics from each chemical-dose regimen pair and chemical-specific parameter estimates from all available data on each chemical. Estimated TK statistics include, but are not limited to, values like the plasma concentration at steady state ($C_{ss}$), maximum observed plasma concentration ($C_{max}$), and area under the plasma concentration vs time curve (AUC). Whereas chemical-specific model parameter estimates include values such as volume of distribution and elimination rate.

**Documenting results with vignettes.** Model evaluation is different from overall package performance evaluation – here we establish if the new model works whereas package performance evaluation ensures we have not broken anything.

Typically, model evaluation requires adding a vignette and new data. Each new model should be evaluated for model accuracy, prior to final incorporation into *httk*, by comparing model predictions against measured data. Statistics, such as the root mean squared $\log_{10}$ error (RMSLE), summarize the concordance between model predictions and data from *in vivo* experiments collected by CvTdb [22]. Model performance analyses and statistics for each model in the *httk* model suite are recorded with "vignettes" [69]. Vignettes generally provide "long-form documentation" with extended examples using functions from the R package [104]. For each manuscript describing an expansion/refinement of the *httk* package we aim to provide a vignette for recreating key figures from that research [8]. Since the vignettes can be rerun with each new package re-build (that is, version), the RMSLE and other performance statistics can be recalculated and may result in new values for one or more of the models.

The vignettes in *httk*, which are written in R markdown documents, provide working code examples for generating figures and performing other analyses using the suite of *httk* models [69]. Including performance calculations in the vignette allows one to monitor the predictive ability of a model with changes in new version releases – including but not limited to new functionalities, bug-fixes, inclusion of new data, etc. For example, if a new functionality introduces errors or inaccuracies, model evaluations within the vignettes provide a mechanism for identifying issues. As *httk* functions are revised and enhanced, older vignettes sometimes become obsolete. Therefore, indicating the *httk* version originally used to develop the vignette or test script is critical for accurate replication of the original analyses and figures. If an archived version of *httk* is needed, it may be obtained at: https://cran.r-project.org/src/contrib/Archive/httk/.

During the package building process, the default is to run all code chunks in every vignette. Some incompatibilities are easily caught in the vignettes during this process. This allows issues to be fixed before submitting the package to CRAN. However, in some cases, time intensive functions and code chunks may cause certain CRAN checks to fail due to time restrictions for vignette compilation. When time intensive examples are necessary, we suggest using the "eval = FALSE" option for these code chunks. This allows users to provide the code for executing an analysis or figure generation but does not execute the computationally intensive code when the running the vignette for testing. To mitigate computational time limits for package rebuilds the beginning of our vignettes typically include an "execute_vignette" argument with a default setting of FALSE. This option tells the vignette not to execute the code chunks within the vignette. However, model developers can set it to TRUE to ensure previous results can be recreated and or to evaluate the potential impacts of *httk* package updates.

When a new vignette is added to the *httk* package it is necessary to include all the necessary data, models, and corresponding functions within the package. All relevant data should be saved as an ".RData" file in the "httk/data" sub-directory of *httk*. Additionally, as with any other data for the package the new data should be documented using *roxygen2* [103] in "httk/R/data.R". Any additional (non-*httk*) functions used in the vignette, which are not part of another existing R package, should be defined within the vignette or in an R script with *roxygen2* documentation [103] and placed in the "httk/R" sub-directory.

The default figure size for vignettes is 3 by 3 (inches), which is quite small but necessary for maintaining a package file size that meets CRAN requirements (that is, 5MB) [105]. Note, if you are using *ggplot2* [106] for figures, then to improve readability, at the default figure size, we suggest commenting out all text size specifications. The *ggplot2* package is generally good at performing the appropriate scaling when rendering figures within a vignette.

**Benchmarking *httk*.** With the addition of many new models and data across *httk* versions it is not only crucial to track model performance over time, but to also track and evaluate the overall performance of the package. A longitudinal evaluation of the package ensures each HTTK model, and any new data included over time, is not degrading the predictive ability of models from previous versions. Recently, the benchmark_httk() function was added to *httk* to provide some of these evaluations and assess the impact of changes over version history – including but not limited to updates in data or code, new models, and new feature implementations – by providing predictive performance benchmarks.

Several possible benchmarks are available via the benchmark_httk() function – including basic performance statistics, Monte Carlo steady state uncertainty estimations, *in vivo* statistics, and tissue partitioning coefficient checks. Table 11 provides additional details about the benchmark(s) returned from each of the checks along with brief descriptions. In addition to obtaining numeric output for each of these outputs, the 'make.plots' argument allows users to visualize the historical performance by generating plots of the performance statistics across version releases to CRAN, including the current version. Logical arguments in the benchmark_httk() function, respective to each of the checks, allow users to control which results are reported. By default, all benchmark checks are returned. We refer readers to the benchmark_httk() help file, for further details on this function and its checks.

Sometimes adding a model may identify a bug in, or a required change to, core *httk* functionality. If a modification to *httk* touches any of the function listed in Table 6 or Table 7 there is a higher level of scrutiny to these code updates/additions. We typically want the default operation to remain the same but explain how user can now tweak that functionality. Therefore, core functionality changes should have a toggle to return to previous behavior, if possible. In addition to using benchmark_httk(), we can build a tarball version of the package (that is, ".tar.gz" file) and use the built-in R package "checks" to look for unintended consequences. Building and checking R packages can be performed from the command shell or by using the R package *devtools* [89]. Running check runs all examples in the code documentation and compares the output of test scripts ("httk/tests") to saved outputs showing expected performance.

For retroactive comparisons, benchmark_httk() performance statistics were obtained from previous versions of *httk* by manually re-installing each archived CRAN release of the package. Historical benchmark statistics are stored in the httk.performance data table, which is included in the *httk* R package via the Tables.RData file. Tracking the benchmark performance statistics allows us to evaluate the overall predictivity of *httk* across released versions. In particular, whether the current version is performing better or worse compared to the previous versions. Furthermore, evaluating benchmark performance statistics as a release preparation step may identify major issues in the package prior to releasing new versions. For

**Table 11. Series of performance benchmarks available through the benchmark_httk() function in httk. These performance statistics are calculated and stored in the httk.performance table for each version release of httk to track performance of the package over time.**

| Benchmark Check | Performance Statistics | Description |
|---|---|---|
| Basic | N.steady.state | Number of chemicals with sufficient data for steady state *in vitro* to *in vivo* (IVIVE) modeling. |
| | calc_analytic.units | Check on whether the conversion between units, mg/L and $\mu$M, for analytic steady state modeling is accurate. Ratio result between mg/L and $\mu$M*(1000/MW) should be 1. |
| | Calc_mc.units | Check on whether the conversion between units, mg/L and $\mu$M, for Monte Carlo steady state uncertainty modeling is accurate. Ratio result between mg/L and $\mu$M*(1000/MW) should be 1. |
| | Solve_pbtk.units | Check on whether the conversion between units, mg/L and $\mu$M, for pbtk modeling is accurate. Ratio result between mg/L and $\mu$M*(1000/MW) should be 1. |
| Monte Carlo Steady state | RMSLE.Wetmore | Root mean squared $\log_{10}$ error between the predicted steady state values from literature values –SimCyp [18] and Wetmore [12], Wetmore [13]. |
| | N.Wetmore | Number of chemicals in the RMSLE.Wetmore comparison. |
| | RMSLE.noMC | Root mean squared $\log_{10}$ error between steady state values predicted from the *httk* functions, that is calc_analytic_css() and calc_css(), without Monte Carlo uncertainty estimation. |
| | N.noMC | Number of chemicals in the RMSLE.noMC comparison. |
| *In vivo* Statistics | RMSLE.InVivoCSS | Root mean squared $\log_{10}$ error between steady state values predicted using the calc_analytic_css() function and *in vivo* estimates of steady state. |
| | N.InVivoCss | Number of chemicals in the RMSLE.InVivoCSS comparison. |

adding new performance data from the latest version of *httk* to the httk.performance dataset, we refer readers to the load_package_data_tables.R in *httk* GitHub repository (https://github.com/USEPA/CompTox-ExpoCast-httk/tree/main/datatables).

## Supporting information

**S1 File. Excel file containing brief descriptions of the components in the modelinfo file and the organization of components by their relevant role in adding/extending a generic HTTK model in *httk*** Note that this file reflects current components and formatting of the modelinfo file as of April 2024 - updates may be necessary for future versions of the package.
(XLSX)

**S2 File. Excel file containing brief descriptions of functions within the R package, which modelers may want to use when integrating/extending their new generic HTTK model into *httk*** The descriptions are not meant to be an exhaustive description of existing functions, which are described by R "help" documentation. Rather this is a reference guide for modelers. Note that this file reflects the current state of the *httk* package and the functions therein as of April 2024 – updates may be necessary for future versions of the package.
(XLSX)

**S3 File. MCSim [86] computer language description of the Linakis [10] model** Provided file "S3_MCSim_example.txt" can be renamed "model_gas_pbtk.model" to match the original. This file does not get stored within the *httk* R package.
(TXT)

**S4 File. C computer language [88] description of the Linakis [10] model as initially generated by MCSim** This "raw" C file is generated from the MCSim model definition file using the MCSim 'mod' function. Provided file "S4_raw_C_example.txt" can be renamed "model_gas_pbtk-raw.c" to match the original. This file does not get stored within any of the sub-directories of the 'httk' R package directory. The file needs to be formatted before incorporation into the R package.
(TXT)

**S5 File. C computer language description of the Linakis [10] model as modified from the "model_gas_pbtk-raw.c" file for integration into** *httk.* **File must be renamed from "S5_C_ for_httk_example.txt" to "model_gas_pbtk.c** ". Once this file is complete it should be stored in the package sub-directory 'httk/src' with the other model C files.
(TXT)

**S6 File. R computer language [107] documentation of the Linakis [10] model for "model aware"** *httk* **functions.** Script contains the definition of the model.list entry for the model "gas_pbtk". File must be renamed from "S6_modelinfo_example.txt" to "modelinfo_gas_pbt-k.R". Once this file is complete it should be stored in the package sub-directory 'httk/R' with other R scripts.
(TXT)

**S7 File. R computer language function for generating chemical-specific parameters for the Linakis [10] model.** File should be renamed from "S7_param_func_example.txt" to "parame-terize_gas_pbtk.R". Once this file is complete it should be stored in the package sub-directory 'httk/R' with other R scripts.
(TXT)

**S8 File. R computer language script containing the function definition for preparing data, solving the Linakis [10] model ODE (by calling the C file), and preparing the output in a user ready format.** File should be renamed from "S8_solve_model_wrapper_example.txt" to "solve_gas_pbtk.R". Once this file is complete it should be stored in the package sub-directory 'httk/R' with other R scripts.
(TXT)

**S9 File. R markdown containing the model evaluation and figure replication code used for Linakis [10].** File should be renamed from "S9_Vh_Linakis2020.txt" to "Vh_Linakis2020. Rmd". Note that there are some updates from the original compilation of the markdown file. Once this file is complete it should be stored in the package sub-directory 'httk/vignettes' with the other vignettes (both general and model evaluation).
(TXT)

## Data availability

There is a fixed set of chemical-specific HTTK parameters distributed with each version of *httk*. Without modifying the chem.physical_and_invitro.table these are all the chemicals that are potentially available to use in a new *httk* model. The *in vitro* measured values can be expanded with structure based predictions using the load_sipes2017() [19], load_pradeep2020() [94], and load_dawson2021() [95] functions in *httk*. Further, only the parameters within chem.physical_and_invitro.table are available for specifying a chemical. Therefore, to develop a new HTTK model that is compatible with chemical-specific parameters available in *httk*, that model must depend on the available parameters. The current set of physico-chemical properties include: $\log_{10}$ membrane affinity (logMA), $\log_{10}$ hydrophobicity (logP), $\log_{10}$ water-air partition coefficient (logPwa), $\log_{10}$ Henry's law coefficient (logHenry), $\log_{10}$ water solubility (logWSol), melting point (MP), molecular weight (MW), ionization equilibria (pKa_Accept and pKa_Donor), and chemical class. Additionally, the existing *in vitro/in vivo* measured properties include: intrinsic hepatic clearance ($Cl_{int}$; Clint), fraction unbound in plasma ($f_{up}$; Funbound.plasma), Caco-2 membrane permeability (Caco2.Pab), fraction absorbed from gut (Fabs), fraction surviving gut metabolism (Fgut), fraction surviving first-pass hepatic metabolism (Fhep), systemic oral bioavailability (Foral), and blood to

plasma ratio (Rblood2plasma). If a new parameter is needed (which has occurred previously) both the chem.physical_and_invitro.table dataset and the relevant core functions for accessing the data must be revised.

## Acknowledgements

The authors thank Drs. Dustin Kapraun, Kelsey Vitense, and Todd Zurlinden for their constructive feedback during the U.S. EPA internal reviews of the manuscript. We appreciate Dr. R. Woodrow Setzer's insights into dynamic modeling in R and statistical analysis of PBTK models. We appreciate programming support from Mohideen Marikar.

## Author contributions

**Conceptualization:** Sarah E. Davidson-Fritz, Gregory S. Honda, Matthew W. Linakis, Robert G. Pearce, Mark A. Sfeir, Michael J. Devito, John F. Wambaugh.

**Data curation:** Sarah E. Davidson-Fritz, Robert G. Pearce, Mark A. Sfeir, John F. Wambaugh.

**Formal analysis:** Sarah E. Davidson-Fritz.

**Funding acquisition:** Michael J. Devito.

**Investigation:** Sarah E. Davidson-Fritz, Caroline L. Ring, Celia M. Schacht, Xiaoqing Chang, Miyuki Breen, Annabel Meade, Robert G. Pearce, James P. Sluka, John F. Wambaugh.

**Methodology:** Caroline L. Ring, Marina V. Evans, Celia M. Schacht, Xiaoqing Chang, Gregory S. Honda, Matthew W. Linakis, Annabel Meade, Robert G. Pearce, John F. Wambaugh.

**Project administration:** Sarah E. Davidson-Fritz, John F. Wambaugh.

**Resources:** Michael J. Devito, John F. Wambaugh.

**Software:** Sarah E. Davidson-Fritz, Caroline L. Ring, Celia M. Schacht, Miyuki Breen, Gregory S. Honda, Matthew W. Linakis, Annabel Meade, Robert G. Pearce, Mark A. Sfeir, John F. Wambaugh.

**Supervision:** Sarah E. Davidson-Fritz, John F. Wambaugh.

**Validation:** Caroline L. Ring, Elaina Kenyon, Robert G. Pearce, John F. Wambaugh.

**Visualization:** Sarah E. Davidson-Fritz, John F. Wambaugh.

**Writing – original draft:** Sarah E. Davidson-Fritz, John F. Wambaugh.

**Writing – review & editing:** Sarah E. Davidson-Fritz, Marina V. Evans, Elaina Kenyon, Matthew W. Linakis, James P. Sluka, Michael J. Devito, John F. Wambaugh.

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
