## [Decision Letter · Decision Letter 0]

23 Dec 2024

PONE-D-24-38287Enabling Transparent Toxicokinetic Modeling for Public Health Risk AssessmentPLOS ONE

Dear Dr. Wambaugh,

Thank you for submitting your manuscript to PLOS ONE. After careful consideration, we feel that it has merit but does not fully meet PLOS ONE’s publication criteria as it currently stands. Therefore, we invite you to submit a revised version of the manuscript that addresses the points raised during the review process.

Thank you for considering our journal for publication of your manuscript. Your manuscript has now been peer-reviewed, could you please take their comments into account and submit a revised version of your manuscript for our consideration?

We look forward to receiving your revised manuscript.

Kind regards,

Antonio Peña-Fernández, PhD

Academic Editor

PLOS ONE

Journal Requirements:

[The United States Environmental Protection Agency (EPA) through its Office of Research and Development (ORD) funded the research described here. This project was supported by appointments to the Internship/Research Participation Program at ORD and administered by the Oak Ridge Institute for Science and Education through an interagency agreement between the U.S. Department of Energy and U.S. EPA. J.P.S acknowledges funding support from the U.S. EPA in grant USEPA RD840027].

Please include this amended Role of Funder statement in your cover letter; we will change the online submission form on your behalf."

3. Please expand the acronym “ U.S. EPA” (as indicated in your financial disclosure) so that it states the name of your funders in full.

4. Thank you for uploading your study's underlying data set. Unfortunately, the repository you have noted in your Data Availability statement does not qualify as an acceptable data repository according to PLOS's standards.

5. Please include captions for your Supporting Information files at the end of your manuscript, and update any in-text citations to match accordingly. Please see our Supporting Information guidelines for more information: http://journals.plos.org/plosone/s/supporting-information .

Additional Editor Comments:

Dear authors,

Thank you for considering our journal for publication of your manuscript. Your manuscript has now been peer-reviewed, could you please take their comments into account and submit a revised version of your manuscript for our consideration?

Best wishes,

Antonio

Reviewers' comments:

Reviewer's Responses to Questions

**Comments to the Author**

1. Is the manuscript technically sound, and do the data support the conclusions?

Reviewer #1: Yes

Reviewer #2: Yes

Reviewer #3: Yes

Reviewer #4: Yes

2. Has the statistical analysis been performed appropriately and rigorously? 

Reviewer #1: Yes

Reviewer #2: Yes

Reviewer #3: Yes

Reviewer #4: Yes

3. Have the authors made all data underlying the findings in their manuscript fully available?

Reviewer #1: Yes

Reviewer #2: No

Reviewer #3: Yes

Reviewer #4: Yes

4. Is the manuscript presented in an intelligible fashion and written in standard English?

Reviewer #1: Yes

Reviewer #2: Yes

Reviewer #3: Yes

Reviewer #4: Yes

5. Review Comments to the Author

Reviewer #1: This manuscript describes the integration and evaluation of new PBTK models into an existing R package (httk) in order to improve in vitro to in vivo extrapolation, which is critical for human health risk assessment. The manuscript is written exceptionally well with clear referals to case studies and model performance evaluations. The manuscript also addresses limitations of the existing package and the improvements that will be implemented by the authors. I do not have any major queries that would impact on the current standard of the manuscript and would therefore suggest acceptance of the manuscript without revision.

Reviewer #2: This manuscript describes the process of adding new TK models to the httk R package. The topic is relevant, particularly given the increasing reliance on in vitro and in silico methods for chemical risk assessment. While the manuscript provides a valuable overview of httk and its functionalities, several major revisions are needed to enhance clarity, accessibility, and impact. Specifically, the manuscript needs to be more concise and focused on the practical aspects of model integration, with less emphasis on background information already available elsewhere. A more concrete and worked-through example is essential for readers to grasp the process.

Major Revisions:

Streamline the Introduction: The introduction is lengthy and contains substantial background on HTTK and risk assessment that could be condensed or removed. Focus on the specific gap this manuscript addresses – the need for clear guidelines on integrating new models into httk – and the benefits of doing so (e.g., community contributions, model standardization, improved reproducibility).

Simplify and Refocus the Results Section: The current results section blends model integration details with evaluation results. Separate these clearly. The re-evaluation of the Linakis et al. model is interesting but distracts from the core purpose of the manuscript. Either move this to a supplementary document or significantly shorten it, focusing on specific changes and their impact on RMSLE. Emphasize the practical implications of observed changes for model developers. The discussion of benchmark_httk() is more appropriate for the methods.

Provide a Concrete, Worked-Through Example: The most significant weakness is the lack of a clear, step-by-step example. While the gas inhalation model is mentioned and supplementary files are provided, the manuscript doesn't guide the reader through the process of integration. Include a simplified, illustrative example within the main text, demonstrating the creation of a basic model (even a one-compartment model) and its integration into httk. Show concrete code snippets and explain the purpose of each step. This example should be the central focus of the manuscript.

Reorganize the Methods Section: The methods section is currently quite dense and difficult to follow. Reorganize it into more logical subsections with clear headings, following the steps of model integration described in Table 1. Each subsection should describe the specific tasks involved in that step, the relevant httk functions and data, and provide short, illustrative code examples where appropriate. Move the discussion of benchmark_httk() from the Results to the "Model Evaluation" subsection of the Methods.

Clarify the Role of MCSim: The manuscript mentions using MCSim for model description but doesn't clearly explain its advantages or provide enough guidance on its use. Expand this discussion and provide a simple MCSim example within the worked-through example mentioned above. If the authors intend to encourage MCSim use for new httk model contributions, this warrants more attention.

Improve Table and Figure Presentation: Several tables are overly complex and could be simplified or broken down. Table 2, in particular, is very dense. Consider splitting it into smaller, more focused tables. Figures 3, 4, and 5 are helpful but could benefit from clearer labels and more concise captions.

Reduce Redundancy: There is some redundancy in the manuscript, particularly regarding the descriptions of core functions and data tables. Streamline these discussions and avoid repeating information already presented in the tables.

Clarify the Intended Audience: Is this manuscript intended for experienced modelers, R users, or a broader audience? Clarify the intended audience and adjust the level of detail and explanation accordingly.

Minor Revisions:

Carefully proofread the manuscript for typos and grammatical errors.

Ensure consistency in terminology and abbreviations.

Provide more context and explanation for equations and mathematical concepts.

Double-check the accuracy of all URLs and references.

By addressing these major revisions, the manuscript can be significantly improved and provide a truly practical and accessible guide for integrating new TK models into httk, fostering community contributions and advancing the field of HTTK modeling

Reviewer #3: - Please remove the quote from the end of the Introduction section. While Dr Feynman is the best, and I absolutely love the quote itself, it seems out of place and unnecessary here.

- A lot of the Methods section feels like it belongs in the Supplementary section

Reviewer #4: In their present work, Sarah et al. describe the integration and evaluation of new physiologically based toxicokinetic (PBTK) models into an open-source R package. The research is robust, the manuscript is well-structured, and the authors have meticulously detailed each step of the model development process. I would like to offer a few suggestions to further enhance the manuscript:

1. Abstract: The abstract is quite lengthy. It would be beneficial to make it more concise while retaining the essential details.

2. It would be helpful if the authors could discuss whether a similar tool is available in other programming languages, such as Python.

3.The resolution of Figure 2 is quite low. Replacing it with a higher-resolution image would significantly improve its clarity and visual appeal.

6. PLOS authors have the option to publish the peer review history of their article (what does this mean? ). If published, this will include your full peer review and any attached files.

**Do you want your identity to be public for this peer review?** For information about this choice, including consent withdrawal, please see our Privacy Policy .

Reviewer #1: No

Reviewer #2: **Yes: ** Xuehai Wang

Reviewer #3: No

Reviewer #4: No

---

## [Author Response · Author response to Decision Letter 1]

6 Feb 2025

Rebuttal Letter PONE-D-24-38287

Enabling Transparent Toxicokinetic Modeling for Public Health Risk Assessment

A rebuttal letter responds to each point raised by the academic editor and reviewer(s).

Associate Editor

Thank you for considering our journal for publication of your manuscript. Your manuscript has now been peer-reviewed, could you please take their comments into account and submit a revised version of your manuscript for our consideration?

Please submit your revised manuscript by Feb 06 2025 11:59PM.

We have revised our manuscript formatting to match these guidelines. In particular we have removed figures from the main document and now provide them as TIFF files.

[The United States Environmental Protection Agency (EPA) through its Office of Research and Development (ORD) funded the research described here. This project was supported by appointments to the Internship/Research Participation Program at ORD and administered by the Oak Ridge Institute for Science and Education through an interagency agreement between the U.S. Department of Energy and U.S. EPA. J.P.S acknowledges funding support from the U.S. EPA in grant USEPA RD840027].

Please include this amended Role of Funder statement in your cover letter; we will change the online submission form on your behalf."

We have added financial disclosures for non-EPA authors. The financial disclosure provided is the standard preferred by the EPA Office of Research and Development. Several of the researchers are within the EPA ORD (the funder). The new funding statement reads:

The United States Environmental Protection Agency (U.S. EPA) through its Office of Research and Development (ORD) funded the research described here. This project was supported by appointments to the Internship/Research Participation Program at ORD and administered by the Oak Ridge Institute for Science and Education through an interagency agreement between the United States Department of Energy and U.S. EPA. JPS acknowledges funding support from the U.S. EPA in grant USEPA RD840027. MWL is a consultant at Ramboll and declares no conflict of interest. XC is an employee of Inotiv, Inc. Her work on this project was funded with federal funds from the National Institute of Environmental Health Sciences, National Institutes of Health under Contract No. HHSN273201500010C.

3. Please expand the acronym “ U.S. EPA” (as indicated in your financial disclosure) so that it states the name of your funders in full.

Done.

4. Thank you for uploading your study's underlying data set. Unfortunately, the repository you have noted in your Data Availability statement does not qualify as an acceptable data repository according to PLOS's standards.

The R package httk is available from Figshare at: https://doi.org/10.23645/epacomptox.6062791.v1

5. Please include captions for your Supporting Information files at the end of your manuscript, and

update any in-text citations to match accordingly. Please see our Supporting Information guidelines for more information: http://journals.plos.org/plosone/s/supporting-information.

We have revised our supporting information to include captions at the end of the manuscript.

Reviewer #1

This manuscript describes the integration and evaluation of new PBTK models into an existing R package (httk) in order to improve in vitro to in vivo extrapolation, which is critical for human health risk assessment. The manuscript is written exceptionally well with clear referals to case studies and model performance evaluations. The manuscript also addresses limitations of the existing package and the improvements that will be implemented by the authors. I do not have any major queries that would impact on the current standard of the manuscript and would therefore suggest acceptance of the manuscript without revision.

Thank you very much for the positive feedback.

Reviewer #2

This manuscript describes the process of adding new TK models to the httk R package. The topic is relevant, particularly given the increasing reliance on in vitro and in silico methods for chemical risk assessment.

• While the manuscript provides a valuable overview of httk and its functionalities, several major revisions are needed to enhance clarity, accessibility, and impact. Specifically, the manuscript needs to be more concise and focused on the practical aspects of model integration, with less emphasis on background information already available elsewhere.

• A more concrete and worked-through example is essential for readers to grasp the process.

Major Revisions

• Streamline the Introduction: The introduction is lengthy and contains substantial background on HTTK and risk assessment that could be condensed or removed. Focus on the specific gap this manuscript addresses – the need for clear guidelines on integrating new models into httk – and the benefits of doing so (e.g., community contributions, model standardization, improved reproducibility).

Thank you for the suggestions. We have shortened the introduction and eliminated redundant information wherever possible throughout the manuscript. Some of the contextual information (that is, the motivation for developing a transparent workflow) has been moved to the discussion.

• Simplify and Refocus the Results Section: The current results section blends model integration details with evaluation results. Separate these clearly. The re-evaluation of the Linakis et al. model is interesting but distracts from the core purpose of the manuscript. Either move this to a supplementary document or significantly shorten it, focusing on specific changes and their impact on RMSLE. Emphasize the practical implications of observed changes for model developers. The discussion of benchmark_httk() is more appropriate for the methods.

• Reorganize the Methods Section: The methods section is currently quite dense and difficult to follow. Reorganize it into more logical subsections with clear headings, following the steps of model integration described in Table 1. Each subsection should describe the specific tasks involved in that step, the relevant httk functions and data, and provide short, illustrative code examples where appropriate. Move the discussion of benchmark_httk() from the Results to the "Model Evaluation" subsection of the Methods.

Potentially divide into 1) Overview Table, 2) Models, Core Fucntionality,

We thank the reviewer for their suggestions. We have jointly addressed these two comments from the reviewer. We agree that there was room for improvement. We have reorganized the Methods and the Results sections to address the reviewer’s concerns about these sections. However, while having considered the reviewer’s recommendations, we propose a slightly different reorganization, since our goal is to help readers address everything that would be needed to get their model incorporated into the public release of httk.

The Methods is now organized into four main sections: 1) Creating and Integrating a New HTTK Model, 2) Core HTTK Functionality, 3) HTTK Data Library, and 4) Tools for HTTK Evaluation. Section one has been rearranged to closely follow the steps of Table 1 for adding a model. Sections 2-4 of the methods provide contextual descriptions of the methods and data that are available to model developers.

The Results section has been similarly reorganized. There are three sections, all of which use the inhalation PBTK model as an example. The first follows Table 1 and describes how a new model was added. A new Table 2 specifically identifies the files created and changes made for that model. Section 2 of the Results describe how the performance of that model was characterized – this is a prerequisite for incorporating any model into the public release of httk. The final section of results described how we determine if integration of the model has impacted performance of the rest of the httk package. We believe that this final section is important as part of the Results because developers of new models will be asked to characterize the impact of adding their model.

• Provide a Concrete, Worked-Through Example: The most significant weakness is the lack of a clear, step-by-step example. While the gas inhalation model is mentioned and supplementary files are provided, the manuscript doesn't guide the reader through the process of integration. Include a simplified, illustrative example within the main text, demonstrating the creation of a basic model (even a one-compartment model) and its integration into httk. Show concrete code snippets and explain the purpose of each step. This example should be the central focus of the manuscript.

Thank you for the suggestions. We have added a new table walking the users through the specific steps and files used to implement the inhalation PBTK model.

• Clarify the Role of MCSim: The manuscript mentions using MCSim for model description but doesn't clearly explain its advantages or provide enough guidance on its use. Expand this discussion and provide a simple MCSim example within the worked-through example mentioned above. If the authors intend to encourage MCSim use for new httk model contributions, this warrants more attention.

The reviewer’s point is well taken. In addition to the new Table 2 (which references the necessary steps including using MCSim), we have added the following discussion of MCSim to the Discussion:

Sophisticated computer programmers are not necessarily expert biological modelers, and vice versa.

There are significant computational advantages to using lower-level model implementations. For example, languages like C or Fortran can be compiled into assembly machine instructions that ran much more rapidly than interpreted languages like R. However, implementing biological models in compilable languages can be tricky. Biological modelers are often more comfortable with descriptive modeling languages which match human expectations (such as ACSL or Berkeley Madonna) rather than languages better suited for rapid computation [38]. Therefore, it is of great use that open source tools like MCSim [39] provide the ability to translate a descriptive language for models (that is, MCSim) into a rapid, compilable (C) language. A public domain guide to MCSim is available at: https://www.gnu.org/software/mcsim/mcsim.html

• Improve Table and Figure Presentation: Several tables are overly complex and could be simplified or broken down. Table 2, in particular, is very dense. Consider splitting it into smaller, more focused tables. Figures 3, 4, and 5 are helpful but could benefit from clearer labels and more concise captions.

We have thoroughly revised former table 2 (now table 3) so that it fits on a single page. We believe it is a critical part of the manuscript. It is intended to cross-walk between how aspects of a model are described in the three languages used by httk. We hesitate to break it into sub-tables because it will not serve its intended purpose if broken up.

We have increased the font sizes in Figure 3 and reduced the length of the captions for Figures 3, 4, and 5.

• Reduce Redundancy: There is some redundancy in the manuscript, particularly regarding the descriptions of core functions and data tables. Streamline these discussions and avoid repeating information already presented in the tables.

Wherever we could, we have reduced duplicative material.

• Clarify the Intended Audience: Is this manuscript intended for experienced modelers, R users, or a broader audience? Clarify the intended audience and adjust the level of detail and explanation accordingly.

We have revised the concluding paragraph of the introduction to hopefully clarify this:

This manuscript is an appeal to the biological modeling community to help expand the physiological and chemical domains addressed by httk. The model development philosophy for httk is create a suite of models, each covering specific exposure scenarios, rather than a single all-encompassing model. Details provided here are intended to aid developers of new models so they may integrate it with httk and use its existing data and functionality.

Minor Revisions:

• Carefully proofread the manuscript for typos and grammatical errors.

Thank you for the suggestion, multiple coauthors have checked for typos and grammar.

• Ensure consistency in terminology and abbreviations.

We have done our best.

• Provide more context and explanation for equations and mathematical concepts.

We have done our best.

• Double-check the accuracy of all URLs and references.

References and URLs have been checked.

• By addressing these major revisions, the manuscript can be significantly improved and provide a truly practical and accessible guide for integrating new TK models into httk, fostering community contributions and advancing the field of HTTK modeling

Reviewer #3

• Please remove the quote from the end of the Introduction section. While Dr Feynman is the best, and I absolutely love the quote itself, it seems out of place and unnecessary here.

Done.

• A lot of the Methods section feels like it belongs in the Supplementary section

While we have extensively reorganized the Methods section and reduced duplicative material to address concerns from other reviewers, we are hesitant to move too much material to the Supplement. This is intended to be a “how to” paper and we hope to provide sufficient detail in the main manuscript.

Reviewer #4

In their present work, Sarah et al. describe the integration and evaluation of new physiologically based toxicokinetic (PBTK) models into an open-source R package. The research is robust, the manuscript is well-structured, and the authors have meticulously detailed each step of the model development process. I would like to offer a few suggestions to further enhance the manuscript:

• Abstract: The abstract is quite lengthy. It would be beneficial to make it more concise while retaining the essential details.

We have rewritten and shortened the abstract

• It would be helpful if the authors could discuss whether a similar tool is available in other programming languages, such as Python.

It is certainly possible to use MCSim with Python (Lohitnavy et al., 2017) and it is also possible to use R package httk through Python using RPy for Python (Kapraun et al., 2017). However, as far as we know the only open source database of chemical-specific HTTK parameters is R package httk. Python package ctx-python (https://github.com/USEPA/ctx-python) allows retrieval of pre-computed predictions made with httk from the CompTox Chemicals Dashboard. We have added this point to the

Discussion:

While there is no reason HTTK data and models must be implemented in R, we are unaware of comparable tools in other, similar languages such as Python. It is certainly possible to use MCSim with Python [79] and it is also possible to use R package httk through

---

## [Editor Report · Decision Letter 1]

9 Feb 2025

PONE-D-24-38287R1Enabling Transparent Toxicokinetic Modeling for Public Health Risk AssessmentPLOS ONE

Dear Dr. Wambaugh,

Thank you for submitting your manuscript to PLOS ONE. After careful consideration, we feel that it has merit but does not fully meet PLOS ONE’s publication criteria as it currently stands. Therefore, we invite you to submit a revised version of the manuscript that addresses the points raised during the review process.

Thank you for submitting a revised version of your manuscript. Please can you re-submit the point by point answer to the reviewers so they can check your modifications?

We look forward to receiving your revised manuscript.

Kind regards,

Antonio Peña-Fernández, PhD

Academic Editor

PLOS ONE

**Additional Editor Comments:**

Dear authors,

Thank you for submitting a revised version of your manuscript. Please can you re-submit the point by point answer to the reviewers so they can check your modifications?

Best wishes,

Antonio

---

## [Author Response · Author response to Decision Letter 2]

10 Feb 2025

Reviewer #1

This manuscript describes the integration and evaluation of new PBTK models into an existing R package (httk) in order to improve in vitro to in vivo extrapolation, which is critical for human health risk assessment. The manuscript is written exceptionally well with clear referals to case studies and model performance evaluations. The manuscript also addresses limitations of the existing package and the improvements that will be implemented by the authors. I do not have any major queries that would impact on the current standard of the manuscript and would therefore suggest acceptance of the manuscript without revision.

Thank you very much for the positive feedback.

Reviewer #2

This manuscript describes the process of adding new TK models to the httk R package. The topic is relevant, particularly given the increasing reliance on in vitro and in silico methods for chemical risk assessment.

• While the manuscript provides a valuable overview of httk and its functionalities, several major revisions are needed to enhance clarity, accessibility, and impact. Specifically, the manuscript needs to be more concise and focused on the practical aspects of model integration, with less emphasis on background information already available elsewhere.

• A more concrete and worked-through example is essential for readers to grasp the process.

Major Revisions

• Streamline the Introduction: The introduction is lengthy and contains substantial background on HTTK and risk assessment that could be condensed or removed. Focus on the specific gap this manuscript addresses – the need for clear guidelines on integrating new models into httk – and the benefits of doing so (e.g., community contributions, model standardization, improved reproducibility).

Thank you for the suggestions. We have shortened the introduction and eliminated redundant information wherever possible. Some of the contextual information (that is, the motivation for developing a transparent workflow) has been moved to the discussion.

• Simplify and Refocus the Results Section: The current results section blends model integration details with evaluation results. Separate these clearly. The re-evaluation of the Linakis et al. model is interesting but distracts from the core purpose of the manuscript. Either move this to a supplementary document or significantly shorten it, focusing on specific changes and their impact on RMSLE. Emphasize the practical implications of observed changes for model developers. The discussion of benchmark_httk() is more appropriate for the methods.

• Reorganize the Methods Section: The methods section is currently quite dense and difficult to follow. Reorganize it into more logical subsections with clear headings, following the steps of model integration described in Table 1. Each subsection should describe the specific tasks involved in that step, the relevant httk functions and data, and provide short, illustrative code examples where appropriate. Move the discussion of benchmark_httk() from the Results to the "Model Evaluation" subsection of the Methods.

Potentially divide into 1) Overview Table, 2) Models, Core Fucntionality,

We have jointly reorganized the Methods and the Results sections to address the reviewer’s concerns about these sections. We thank the reviewer for their suggestions. We agree that there was room for improvement. However, while having considered the reviewer’s recommendations, we propose a slightly different reorganization. Our goal is to help readers address everything that would be needed to get their model incorporated into the public release of httk.

The Methods is now organized into four main sections 1) Creating and Integrating a New HTTK Model, 2) Core HTTK Functionality, 3) HTTK Data Library, and 4) Tools for HTTK Evaluation. Section one has been rearranged to closely follow the steps of Table 1 for adding a model. Sections 2-4 of the methods provide descriptions of the methods and data that are available to model developers.

The Results section has been similarly reorganized. There are three sections all using the inhalation PBTK model as an example. The first follows Table 1 and describes how a new model was added. A new Table 2 specifically references the files created and changes made for that model. Section 2 of the Results describe how the performance of that model was characterized – this is a prerequisite for incorporating any model into the public release of httk. The final section of results described how we determine if integration of the model has impacted performance of the rest of the httk package. We believe that this final section is important as part of the Results because developers of new models will be asked to characterize the impact of adding their model.

• Provide a Concrete, Worked-Through Example: The most significant weakness is the lack of a clear, step-by-step example. While the gas inhalation model is mentioned and supplementary files are provided, the manuscript doesn't guide the reader through the process of integration. Include a simplified, illustrative example within the main text, demonstrating the creation of a basic model (even a one-compartment model) and its integration into httk. Show concrete code snippets and explain the purpose of each step. This example should be the central focus of the manuscript.

Thank you for the suggestions. We have added a new table walking the users through the specific steps and files used to implement the inhaltion PBTK model.

• Clarify the Role of MCSim: The manuscript mentions using MCSim for model description but doesn't clearly explain its advantages or provide enough guidance on its use. Expand this discussion and provide a simple MCSim example within the worked-through example mentioned above. If the authors intend to encourage MCSim use for new httk model contributions, this warrants more attention.

The reviewer’s point is well taken. In addition to the new Table 2 (which references the necessary steps including using MCSim), we have added the following discussion of MCSim to the Discussion:

Sophisticated computer programmers are not necessarily expert biological modelers, and vice versa. There are significant computational advantages to using lower-level model implementations. For example, languages like C or Fortran can be compiled into assembly machine instructions that ran much more rapidly than interpreted languages like R. However, implementing biological models in compilable languages can be tricky. Biological modelers are often more comfortable with descriptive modeling languages which match human expectations (such as ACSL or Berkeley Madonna) rather than languages better suited for rapid computation [38]. Therefore, it is of great use that open source tools like MCsim [39] provide the ability to translate a descriptive language for models (that is, MCSim) into a rapid, compilable (C) language. A public domain guide to MCSim is available at: https://www.gnu.org/software/mcsim/mcsim.html

• Improve Table and Figure Presentation: Several tables are overly complex and could be simplified or broken down. Table 2, in particular, is very dense. Consider splitting it into smaller, more focused tables. Figures 3, 4, and 5 are helpful but could benefit from clearer labels and more concise captions.

We have thoroughly revised former table 2 (now table 3) so that it fits on a single page. We believe it is a critical part of the manuscript. It is intended to cross-walk between how aspects of a model are described in the three languages used by httk. We hesitate to break it into sub-tables because it will not serve its intended purpose if broken up.

We have increased the font sizes in Figure 3 and reduced the captions for Figures 3, 4, and 5.

• Reduce Redundancy: There is some redundancy in the manuscript, particularly regarding the descriptions of core functions and data tables. Streamline these discussions and avoid repeating information already presented in the tables.

Wherever we could, we have reduced duplicative material.

• Clarify the Intended Audience: Is this manuscript intended for experienced modelers, R users, or a broader audience? Clarify the intended audience and adjust the level of detail and explanation accordingly.

We have revised the concluding paragraph of the introduction to hopefully clarify this:

We are inviting the biological modeling community to help expand the physiology and chemical domains addressed by httk. This work is intended to help the developers of new models use the existing HTTK data and functionality.

Minor Revisions:

• Carefully proofread the manuscript for typos and grammatical errors.

Thank you for the suggestions, multiple coauthors have checked for typos and grammar.

• Ensure consistency in terminology and abbreviations.

We have done our best.

• Provide more context and explanation for equations and mathematical concepts.

We have done our best.

• Double-check the accuracy of all URLs and references.

References and URLs have been checked.

• By addressing these major revisions, the manuscript can be significantly improved and provide a truly practical and accessible guide for integrating new TK models into httk, fostering community contributions and advancing the field of HTTK modeling

Thank you.

Reviewer #3

• Please remove the quote from the end of the Introduction section. While Dr Feynman is the best, and I absolutely love the quote itself, it seems out of place and unnecessary here.

Done.

• A lot of the Methods section feels like it belongs in the Supplementary section

While we have extensively reorganized the Methods section and reduced duplicative material to address concerns from other reviewers, we are hesitant to move too much material to the Supplement. This is intended to be a “how to” paper and we hope to provide sufficient detail in the main manuscript.

Reviewer #4

In their present work, Sarah et al. describe the integration and evaluation of new physiologically based toxicokinetic (PBTK) models into an open-source R package. The research is robust, the manuscript is well-structured, and the authors have meticulously detailed each step of the model development process. I would like to offer a few suggestions to further enhance the manuscript:

• Abstract: The abstract is quite lengthy. It would be beneficial to make it more concise while retaining the essential details.

We have rewritten and shortened the abstract.

• It would be helpful if the authors could discuss whether a similar tool is available in other programming languages, such as Python.

It is certainly possible to use MCSim with Python (Lohitnavy et al., 2017) and it is also possible to use R package httk through Python using RPy for Python (Kapraun et al., 2017). However, as far as we know the only open source database of chemical-specific HTTK parameters is R package httk. Python package ctx-python (https://github.com/USEPA/ctx-python) allows retrieval of pre-computed predictions made with httk from the CompTox Chemicals Dashboard. We have added this point to the Discussion:

While there is no reason HTTK data and models must be implemented in R, we are unaware of comparable tools in other, similar languages such as Python. It is certainly possible to use MCSim with Python [79] and it is also possible to use R package httk through Python using rpy2 [80] for Python [81]. However, as far as we know the only open-source database of chemical-specific HTTK parameters and models is R package httk. Python package ctx-python (https://github.com/USEPA/ctx-python) allows retrieval of pre-computed predictions made with httk from the CompTox Chemicals Dashboard [63]. R package ctxR [82] provides functionality similar to ctx-python for R.

• The resolution of Figure 2 is quite low. Replacing it with a higher-resolution image would significantly improve its clarity and visual appeal.

We have added a standalone copy of Figure 2 to the submission. The image is saved as a tiff file that we can provide to the journal at arbitrary resolution. We have doubled the resolution to 1200 x 800, but for the time being the version that builds in the PDF may show at lower than optimal resolution.

---

## [Decision Letter · Decision Letter 2]

4 Mar 2025

Enabling Transparent Toxicokinetic Modeling for Public Health Risk Assessment

PONE-D-24-38287R2

Dear Dr. Wambaugh,

We’re pleased to inform you that your manuscript has been judged scientifically suitable for publication and will be formally accepted for publication once it meets all outstanding technical requirements.

Kind regards,

Antonio Peña-Fernández, PhD

Academic Editor

PLOS ONE

Additional Editor Comments (optional):

Dear authors,

Your revised manuscript has been peer-reviewed. The reviewers are satisfied with the changes you have made and I am recommending publication in our journal.

Congratulations,

Antonio

Reviewers' comments:

Reviewer's Responses to Questions

**Comments to the Author**

1. If the authors have adequately addressed your comments raised in a previous round of review and you feel that this manuscript is now acceptable for publication, you may indicate that here to bypass the “Comments to the Author” section, enter your conflict of interest statement in the “Confidential to Editor” section, and submit your "Accept" recommendation.

Reviewer #2: All comments have been addressed

Reviewer #4: All comments have been addressed

2. Is the manuscript technically sound, and do the data support the conclusions?

Reviewer #2: Yes

Reviewer #4: Yes

3. Has the statistical analysis been performed appropriately and rigorously? 

Reviewer #2: Yes

Reviewer #4: Yes

4. Have the authors made all data underlying the findings in their manuscript fully available?

Reviewer #2: Yes

Reviewer #4: Yes

5. Is the manuscript presented in an intelligible fashion and written in standard English?

Reviewer #2: Yes

Reviewer #4: Yes

6. Review Comments to the Author

Reviewer #2: I believe that the revised manuscript addresses all the concerns raised by the reviewers and has been significantly improved.

Reviewer #4: The authors have incorporated the suggested changes. Therefore, I recommend accepting the manuscript in its current form.

7. PLOS authors have the option to publish the peer review history of their article (what does this mean? ). If published, this will include your full peer review and any attached files.

**Do you want your identity to be public for this peer review?** For information about this choice, including consent withdrawal, please see our Privacy Policy .

Reviewer #2: **Yes: ** Xuehai Wang

Reviewer #4: No

---

## [Editor Report · Acceptance letter]

PONE-D-24-38287R2

PLOS ONE

Dear Dr. Wambaugh,

I'm pleased to inform you that your manuscript has been deemed suitable for publication in PLOS ONE. Congratulations! Your manuscript is now being handed over to our production team.

Kind regards,

on behalf of

Dr. Antonio Peña-Fernández

Academic Editor

PLOS ONE